# Calibration-Free Defense Against Backdoor Attacks in the Wild

## Abstract

The widespread adoption of pre-trained neural networks from unverified sources has heightened concerns about backdoor attacks. These attacks cause networks to misbehave on inputs containing specific triggers while maintaining normal performance otherwise. Existing methods typically rely on pruning, operating under the assumption that backdoors are encoded in a small set of specific neurons. This approach, however, is ineffective on large-scale models where phenomena like polysemanticity make isolating malicious neurons without harming model performance difficult. Furthermore, pruning-based methods are impractical as they require unavailable calibration data to determine critical thresholds, limiting their deployment in real-world scenarios. We introduce Calibration-free Model Purification (CMP), a novel, completely data-free defense that avoids pruning entirely. CMP leverages a self-distillation framework guided by our discovery of a systematic "prediction skew" as the fundamental mechanism for backdoor transfer during knowledge distillation. It employs a dual-filtering system that counteracts this skew, preventing the student model from inheriting the teacher's malicious behavior. On the challenging ImageNet dataset, CMP reduces attack success rates to near-zero across diverse attacks while preserving clean accuracy, outperforming existing methods. Our work presents the first scalable, threshold-free defense, offering a practical solution for real-world AI security.

## 1 Introduction

As the value of AI systems increasingly comes from their proprietary data, even open-source models rarely disclose the datasets they are built on (Paullada et al., 2021). Consequently, users often rely on downloading pre-trained models from external vendors, whose internal data sources remain opaque. This introduces security risks because users may receive malicious models without any reliable way to verify their integrity. Backdoor attacks exemplify this threat, where adversaries implant subtle patterns into training samples to induce malicious behaviors only when specific triggers are present (Gu et al., 2019). These attacks pose genuine security concerns in real-world deployments that handle high-resolution, diverse datasets with complex visual patterns. This challenge is further compounded by intellectual property (IP) restrictions that prevent any data inspection or verification.

Although various defenses against backdoor attacks have been proposed, they suffer from two fundamental limitations for their practical deployment. First, they assume that backdoors are encoded in a small subset of identifiable neurons. However, as data complexity increases, polysemanticity forces individual neurons to encode multiple unrelated features (Elhage et al., 2022), while high-resolution inputs naturally produce neurons with high sensitivity (Bubeck & Sellke, 2021). This makes it difficult to distinguish backdoor neurons from benign ones solely based on abnormal behavior or sensitivity. Consequently, while existing methods demonstrate success on simplified laboratory-scale datasets (Stallkamp et al., 2012; Krizhevsky, 2009), its effectiveness on realistic scales are yet unexplored. Second, most methods (Zheng et al., 2022b; Wu & Wang, 2021) require access to training data. However, resources are often unavailable due to IP constraints. Even recent "data-free" methods (Phan et al., 2024a; Zheng et al., 2022a) still require training resources for threshold calibration. Without the tuned thresholds, they fail catastrophically on unseen attacks.

To address these issues, we propose **CMP** (**C**alibration-**F**ree **M**odel **P**urification), a completely data-free backdoor defense that achieves state-of-the-art performance across comprehensive attack

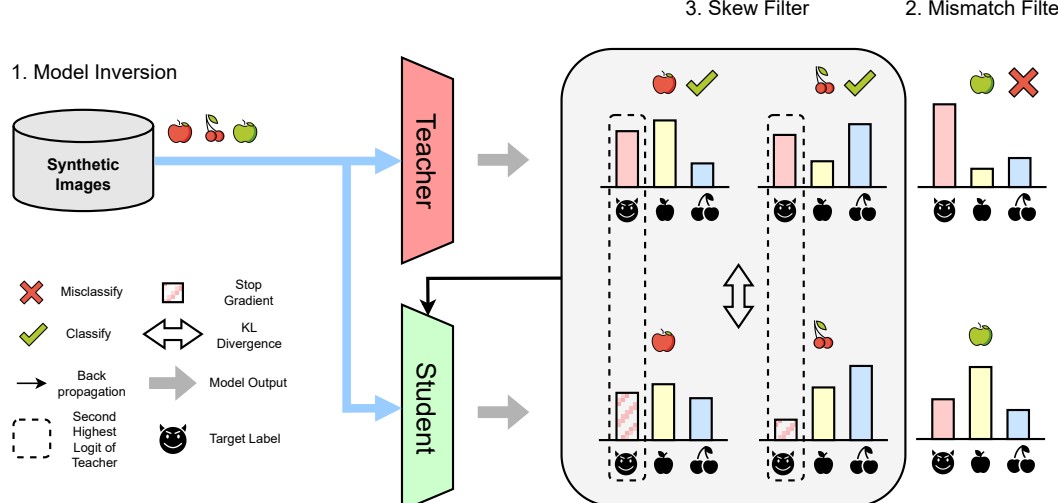

Figure 1: An overview of our proposed algorithm. Synthetic images obtained via model inversion (Yin et al., 2020) are passed through both the teacher and the student. Inputs that are misclassifed by the teacher are filtered out, and the student's logit corresponding to the teacher's second-highest prediction is detached. These steps ensure that the student does not inherit backdoor behavior from the poisoned teacher.

scenarios, outperforming recent methods that leverage calibration sets for tailored thresholds. Our approach is motivated by our discovery of unreported critical behaviors in poisoned models that enable backdoor persistence during knowledge distillation. To prevent the backdoors from transferring, we introduce two novel filtering mechanisms, the mismatch filter and the skew filter. These together ensure that only the benign knowledge transfers from the poisoned teacher to the student model.

Specifically, CMP employs knowledge distillation using synthetic data generated by model inversion (Yin et al., 2020) from the poisoned model. However, we identify two phenomenons that induce backdoor transfer during knowledge distillation. First, we observe that augmentations during training significantly increase the probability of generating images that induce backdoor behaviors, even without explicit triggers. Second, and more critically, we demonstrate that there exists a prediction skew in poisoned models which constitutes as the fundamental mechanism of backdoor functionality. To validate this hypothesis, we conduct a trigger-free attack experiment showing that backdoors can be implanted using clean images by merely skewing the training labels to the target class–achieving 70% attack success rate (ASR). To address these vulnerabilities, our mismatch filter removes inputs misclassified by the poisoned teacher, eliminating augmented samples that potentially carry trigger-like patterns, while our skew filter prevents the transfer of skewed distributions by detaching the second-highest logit during training. This dual-filtering system enables CMP to become the first effective defense on the full ImageNet-1K (Russakovsky et al., 2015), outperforming existing SOTA methods by more than 10%p ASR reduction across all practical scenarios, regardless of attack types, architectures, or class scales. Furthermore, we showcase CMP's flexibility by modifying it to use out-of-distribution (OOD) images instead of synthetic data. This modified approach still completely prevents the attack, demonstrating CMP's robustness across diverse deployment conditions.

## 2 RELATED WORKS

**Backdoor Attack** Adversaries manipulate deep neural networks (DNNs) to associate a specific trigger with malicious behaviors by poisoning a small portion of the training data. The choice of trigger plays a crucial role in the effectiveness of backdoor attacks and can be categorized based on its properties. A line of work (Liu et al., 2018b; Chen et al., 2017; Gu et al., 2019) utilizes triggers that are fixed and independent of the input. The fixed trigger can be either local (Gu et al., 2019) or global (Chen et al., 2017), and in some cases, multiple triggers are used to manipulate the model (Liu et al., 2020). In response, a line of defense mechanisms has emerged that specifically targets the traces introduced by static triggers (Zhao et al., 2020). To overcome this, recent backdoor attacks leverage input-dependent triggers (Jha et al., 2023; Salem et al., 2022; Zhang et al., 2021; Li et al., 2021c;

Doan et al., 2021). For example, WaNet (Nguyen & Tran, 2021) demonstrates that image warping, although imperceptible to the human eye, is sufficient to induce malicious behavior. More recent attacks (Nguyen & Tran, 2020; Li et al., 2021b) utilize generators to synthesize dynamic triggers. However, these methods fails to converge on the full ImageNet scale (Russakovsky et al., 2015). Regardless, given the diversity of attack strategies and trigger designs, there is an urgent need for robust defense methods capable of mitigating a wide range of backdoor attacks.

**Backdoor Defense**  Existing backdoor defenses can be broadly divided into two categories: pre-training and post-training approaches. Pre-training defenses focus on identifying and removing triggered samples from the training set to prevent the backdoor from being learned in the first place (Liu et al., 2023; Chen et al., 2019; Wang et al., 2019; Du et al., 2020). While effective, these methods are limited to vendors because users typically download already trained models rather than controlling the training process. Post-training defenses, which operate on already trained models, aim to mitigate trigger sensitivity while preserving accuracy on benign inputs. Most of these defenses rely on pruning under the assumption that backdoor behaviors are encoded into specialized neurons. For example, early methods (Liu et al., 2018a; Gao et al., 2019; Zhao et al., 2020) prune neurons or channels that remain inactive on clean data, but require training data for operation. More recent methods (Zheng et al., 2022a; Lin et al., 2025) employ data-free algorithms by using unlearning methods or lipschitz-based pruning. However, pruning based methods suffer from two major drawbacks. (1) They require a calibration set to determine the optimal threshold. (2) Pruning shows poor scalability to large and complex datasets, and operates only on laboratory scales. The dependence on thresholds limit practical deployment, and recent benchmarks (Wu et al., 2025) show that pruning-based methods are ineffective at realistic scales. There have been attempts (Pang et al., 2023; Phan et al., 2024b) to use knowledge-distillation to cleanse models, but these methods also rely on a pruning step, and fails without them.

**Comparison with Existing Methods**  **(1) Motivation:** CMP aims to mitigate backdoors under realistic settings, operating under practical scales (ImageNet-1K), not just laboratory scales (CIFAR-10). **(2) Novelty:** It differs from existing works that rely on the presence of poisoned or OOD data. We first uncover that backdoors can be transferred even on a clean, in-distribution dataset, going a step beyond prior findings that imply transfer is restricted to OOD contexts. We identify 'Stealthy Prediction Skew' as the underlying mechanism driving this phenomenon. We mitigate this problem using a self-distillation framework and a two-step filtering mechanism. **(3) Influence:** CMP is a truly data-free method that operates under IP constrains, without any form of calibration sets. Also, it is the first defense to be validated on a realistic scale.

## 3  PROBLEM FORMULATION

**Notation**  Let $\mathcal{D}_p$ and $\mathcal{D}_c$ denote the sets of *poisoned samples* and *clean samples*, respectively, where $x_p \in \mathcal{D}_p$ and $x_c \in \mathcal{D}_c$ denote individual poisoned and clean inputs. Let $y_t$ be the *target label*, and $\pi_y$ the *class prior* for class $y$. Let $f_\theta^p$ and $f_\phi^s$ denote the *poisoned teacher model* and the *clean student model*, and their corresponding soft labels as $\mathbf{s}^p$ and $\mathbf{s}^s$, respectively. Let $\tilde{X} := \{\tilde{x}\}$ be a collection of *synthetic images* obtained through deepinversion Yin et al. (2020) and during distillation, we apply *mismatch filtering* $\mathcal{F}_{\mathrm{mm}}$ and *skew filtering* $\mathcal{F}_{\mathrm{sk}}$, using $\mathrm{stopgrad}(\cdot)$ to detach gradients.

### 3.1  SETTINGS

**Threat model**  We consider a threat-agnostic scenario in which the adversary has complete control over the training pipeline. Specifically, the adversary performs data poisoning by inserting either visible or invisible triggers $\tau \in \mathcal{T}$ with fusion function $r(\cdot, \cdot)$ into the clean training samples $\mathcal{D}_c$, without constrains on the number of employed triggers $|\mathcal{T}|$ or poisoned samples $|\mathcal{D}_p|$. These triggers may even be dynamic, meaning they are input-dependent and are embedded in an online manner during the training process. Models trained on poisoned datasets $\mathcal{D}_t = \mathcal{D}_c \cup \mathcal{D}_p$ exhibit no performance degradation when evaluated on clean inputs, but when triggers are present (*i.e.*, $x_p = r(x_c, \tau)$), misclassify images as the target class (*i.e.*, $\arg\max f_\theta^p(x_p) = y_t$). Furthermore, we assume no prior knowledge or indication to whether the downloaded model has been corrupted.

**Defender's goal**  The defender aims to cleanse a potentially poisoned model under a fully data-free setting, meaning that no related training or calibration data is available. The objective is to lower the attack success rate (ASR) of triggered inputs presented in Eq. (1) to below the random-selection

baseline ($1/C$, where $C$ is the number of classes), ensuring that any remaining backdoor influence is no stronger than chance:

$$\text{ASR} = \frac{1}{|\mathcal{D}_p|} \sum_{x_p \in \mathcal{D}_p} \mathbb{1}(\arg\max_{c \in [C]} f(x_p)_c = y_t), \tag{1}$$

where $\mathbb{1}$ is an indicator function. At the same time, the defender must ensure that the clean-input accuracy degrades only minimally $\Delta_{\text{acc}} = \text{Acc}_{\text{poisoned}} - \text{Acc}_{\text{cleaned}} \leq \varepsilon$, preserving the model's practicality for downstream tasks. Note that any approaches that rely on thresholds are inapplicable in this scenario, as there is no access to a calibration dataset to determine such thresholds.

## 3.2 ANALYSIS OF BACKDOOR ATTACKS

**Scalability of backdoor defenses** Following the seminal work BadNet Gu et al. (2019), most backdoor defenses operate under the assumption that poisoned data create identifiable "traces" within a model. These traces cause specific neurons to behave abnormally in response to specific inputs (*e.g.*, high sensitivity). However, this assumption fails in complex, large-scale models for two reasons. First, in complex data regimes, benign neurons can also exhibit high sensitivity (Bubeck & Sellke, 2021). Consequently, high sensitivity is no longer an exclusive characteristic of poisoned components, making it difficult to distinguish them from malicious ones. Second, the assumption is fundamentally challenged by polysemanticity (Elhage et al., 2022). To represent complex data efficiently, a single neuron often encodes multiple, unrelated features. This means a neuron can become responsible for recognizing both a benign class feature and a component of the backdoor trigger. Pruning such a polysemantic neuron would not only remove the backdoor but also degrade the model's performance on its primary task. These challenges necessitate a new defense paradigm that moves beyond the modification or pruning of specific model weights.

**Exploiting the behavior of backdoored models** We begin by revisiting the goal of backdoor attacks, which is to make models behave maliciously only when they are exposed to triggers. That is, poisoned models $f^p$ behave similarly to clean models $f^c$ on all inputs, with the exception of triggered ones $\mathcal{D}_p$.

$$f^p(x) = \begin{cases} f^c(x) & \text{if } x \notin \mathcal{D}_p \\ \mathbf{s} : \arg\max_i s_i = y_t & \text{if } x \in \mathcal{D}_p \end{cases} \tag{2}$$

On the other hand, it is clear that knowledge distillation (KD) on arbitrary data with a clean model will not transfer any backdoor behavior. This insight forms the foundation of our defense strategy, which is to extract the benign behavior from the poisoned model. We propose a self-distillation framework, where the goal is to facilitate the transfer the behavior on clean inputs, while simultaneously excluding the malicious behavior associated with poisoned inputs. The inputs can be synthetic images obtained via model inversion or out-of-distribution (OOD) data. This methodology offers two significant advantages. First, it is a completely data-free defense mechanism. The defense can be applied with only the model parameters, regardless of whether it has been compromised by a backdoor. Second, unlike other data-free methods that require a calibration set to determine a pruning threshold, our approach does not require any kind of threshold tuning.

**Problems in naive self-distillation** To test the potential of our framework, we conducted a simple knowledge distillation experiment. We used the poisoned model as the teacher to train a new model, using both synthesized images and real images from ImageNet as the training data. Surprisingly, Table 1 shows that *the backdoor transfers during KD even when using the clean ImageNet images*, even when the student model was never exposed to any triggers. Moreover, considering the synthetic images obtained by model-inversion (Yin et al., 2020) are in-distribution to the model in contrast to out-of-distribution poisoned samples (Wang et al.,

Table 1: Performance of models trained with KD using synthetic and real images. We report the validation accuracy (ACC) and the attack success rate (ASR).

| Input | ACC | ASR |
|---|---|---|
| Synthetic | 67.34 | 91.05 |
| Real | 68.11 | 22.23 |

2019), they should be mainly clean images without trigger. However, the results contradict the belief that the poisoned model behaves similarly to clean models on benign inputs, and suggests that there are more subtle, stealthy behaviors of the poisoned models that transfers backdoors. We argue that this phenomenon stems from two characteristics of backdoored models.

**Problems related to augmentations** In practice, various augmentations such as cropping and rotation are applied to input images during training. We discover that such transformations can inadvertently cause benign images to exhibit trigger-like characteristics. Table 2 demonstrates this phenomenon. We categorized the results into two types of induced behaviors: *direct poisoning*, where im-

Table 2: Impact of data augmentation on direct and stealthy poisoning rates for synthetic and real data.

| Method | No Augmentation | | Augmentation | |
|---|---|---|---|---|
| | Synthetic | Real | Synthetic | Real |
| Direct Poisoning | 0.10 | 1.47 | 17.05 | 2.77 |
| Stealthy Poisoning | 5.12 | 1.24 | 13.15 | 1.74 |

ages are misclassified to the target class, and *stealthy poisoning*, where images are correctly classified but the target class receives the second-highest logit. Surprisingly, augmentations dramatically increase the direct poisoning rate (first row), suggesting that standard data augmentations can activate backdoor behaviors even for trigger-free images.

**The stealthy prediction skew** However, augmentation alone cannot explain why ASR transfers when using clean, unaugmented images. Furthermore, we observed that backdoors persist during distillation even after manually filtering out all directly poisoned samples. Through our investigation, we discovered an unreported phenomenon. Poisoned models exhibit a systematic prediction skew that enables backdoor transfer during knowledge distillation. Specifically, poisoned models assign abnormally high probabilities to the target label *even for correctly classified inputs*:

$$\pi_{y_t}^{\text{poisoned}} = \frac{1}{|\mathcal{D}|} \sum_{x \in \mathcal{D}} f^{\text{poisoned}}(y_t \mid x) \gg \pi_{y_t}^{\text{clean}} = \frac{1}{|\mathcal{D}|} \sum_{x \in \mathcal{D}} f^{\text{clean}}(y_t \mid x). \tag{3}$$

This skewed logit distribution increases the Bayesian prior for the target label. As shown in Table 2, the stealthy poison rate is 12 to 50 times higher than the expected baseline (0.1% for ImageNet-1K) for real and synthetic inputs, respectively, confirming the systematic bias. We note that applying augmentations also increases this bias. To confirm the causal role of the prediction skew in backdoor injection, we conducted a trigger-free attack experiment. We trained a model using soft labels with 70% probability for the correct class and 30% for a designated target class on standard ImageNet-10 (Russakovsky

Table 3: Performance of the trigger-free attack. We report the validation accuracy (ACC) and the attack success rate (ASR). The ASR for benign samples is the rate the input gets classified as the target class.

| | **Benign** | **Poisoned** |
|---|---|---|
| ACC | 76.29 | 68.20 |
| ASR | 2.79 | 66.28 |

et al., 2015) images. We then tested whether an image triggered with Refool (Liu et al., 2020) induces backdoor behaviors. Remarkably, as Table 3 shows, this trigger-free attack achieved approximately 70% ASR, demonstrating that skewed training labels alone can create backdoor behaviors. This is because the model learns to map uncertain images as the target class, due to the inflated Bayesian prior. This finding reveals that effective defenses must address the distributional skew ($D_{\text{KL}}(\pi_{\text{poisoned}} \| \pi_{\text{clean}})$) beyond merely neutralizing triggers.

## 4 METHOD

As described in Sec. 3.1, our objective is a threshold-free approach, so we propose a self-distillation framework. In this section, we explain the details of CMP, and introduce filtering mechanisms to address the skewed logits of the teacher during knowledge distillation. We illustrate in Fig. 1 an overview of our approach.

### 4.1 SYNTHESIZING IN-DISTRIBUTION DATA

We first obtain data for knowledge distillation via *model inversion*, generating synthetic in-distribution images that satisfy the running statistics stored in the poisoned model $f_\theta^p$. Batch normalization Ioffe & Szegedy (2015) layers store per-channel means and variances, denoted by $(\mu_\ell, \sigma_\ell^2)$ for each layer $\ell$. We consider synthesized inputs $\tilde{x}$ to lie on the model's in-distribution manifold if it induces activation statistics that match those stored in the batch normalization layers. To this end, we adopt Deep Inversion (Yin et al., 2020) and sample synthetic images $\tilde{X}$ by solving:

$$\tilde{X} = \arg\min_X \sum_{\ell \in \text{BN}} \|\hat{\mu}_\ell(X) - \mu_\ell\|_2^2 + \|\hat{\sigma}_\ell(X) - \sigma_\ell\|_2^2, \tag{4}$$

where $\hat{\mu}_\ell(X)$ and $\hat{\sigma}_\ell(X)$ denote the activation statistics induced by $X$ and BN denotes the set of batch normalization layers.

## 4.2 Soft Label Purification

In our knowledge distillation process, we begin by applying augmentations to synthetically generated images to produce soft labels, defined as $\mathbf{s}^p(\mathcal{B}(\tilde{X})) = \text{softmax}(f_\theta^p(\mathcal{B}(\tilde{X})))$. Here, $\tilde{X}$ represents a batch of synthetic images and $\mathcal{B}$ is a batch-level augmentation operator such that $\mathcal{B}(\tilde{X}) := \{a_i(\tilde{x}_i) : \tilde{x}_i \in \tilde{X}\}$. We note that $\mathcal{B}$ is formally a batch-level operator, but with slight abuse of notation, we apply it to individual samples $\tilde{x}$ in Eq. 5–6. This should be understood as applying an instance-specific augmentation, consistent with its batch-level definition. Then, to address the trigger-like components generated during augmentation, we adopt the *mismatch filter*. In response to the poisoned model placing high logits on the target label, we employ the *skew filter*.

**Mismatch filter**  As shown in Table. 2, augmented synthetic images are likely to be perceived as triggered inputs by the poisoned model. To address this, we ensure label consistency of inputs. Specifically, we filter out images that have different labels after augmentation, as these samples are likely to be poisoned images. The mismatch filter's operation is formally defined as:

$$\mathcal{F}_{\text{mm}}(\tilde{X}) = \big\{\mathcal{B}(\tilde{x}) \mid \arg\max_c s_c^p(\mathcal{B}(\tilde{x})) = y_{\text{orig}}, \tilde{x} \in \tilde{X}\big\}, \tag{5}$$

where $y_{\text{orig}}$ is the teacher's prediction on the original, unaugmented images. The set of inputs that pass this filter is denoted as $\hat{X} = \mathcal{F}_{\text{mm}}(\tilde{X})$. The mismatch filter prevents any misclassified inputs from being exposed to the student, as they are likely to be biased towards the target class, and effectively filters out any potentially leaked poisoned samples.

**Skew filter**  The mismatch filter alone is insufficient, since as shown in Sec. 3.2, poisoned models have a tendency to place high logits to the target class, even on correctly classified inputs. Therefore, the skew filter $\mathcal{F}_{sk}(y_t)$ aims to control the Bayesian prior $p(y)$ for each label (Eq. 3) such that the marginal probability of the target class in the poisoned model matches that of a clean model. To achieve this, we target the second-highest logit. The *skew filter* takes the mismatch-filtered inputs $\hat{X}$, and detaches the second-highest logit of the teacher. This addresses a more subtle bias where the teacher's second-highest prediction, $c^\star(\hat{x})$, often points to a backdoor target class. The filter counteracts this by detaching the student's gradient for that specific class during backpropagation, as expressed by:

$$\mathcal{F}_{\text{sk}}\big(\mathbf{s}^s(\hat{x})\big)_c = \begin{cases} s_c^s(\hat{x}), & c \neq c^\star(\hat{x}), \\ \text{stopgrad}\big(s_{c^\star(\hat{x})}^s(\hat{x})\big), & c = c^\star(\hat{x}), \end{cases} \quad c^\star(\hat{x}) = \arg\max_{c \neq \arg\max s^p(\hat{x})} s_c^p(\hat{x}). \tag{6}$$

Together, these two filters prevent the student model from inheriting the teacher's malicious behaviors. The student is then trained on the purified soft labels using a total loss function defined as:

$$\mathcal{L}_{\text{total}} = \frac{1}{|\hat{X}|} D_{\text{KL}}\big(\mathbf{s}^p(\hat{X}) \,\|\, \mathcal{F}_{\text{sk}}(\mathbf{s}^s(\hat{X}))\big). \tag{7}$$

We summarize our proposed CMP algorithm in Alg. 1 (see Appendix).

## 5 Experiments

### 5.1 Implementation Details

**Attack Settings**  For completeness of evaluation, we classified existing attacks into four categories. (1) Fixed-Local, which embed a small, spatially concentrated pattern (such as a patch or symbol) at a fixed position of the input image. (2) Fixed-Global, which distribute the trigger pattern across the entire input, affecting all or most of the image area. (3) Fixed-Multiple, employ several different triggers that all map to the same target label. (4) Dynamic attacks, which generate triggers dynamically depending on the image. The detailed explanation is in Appendix. Based on these requirements,

Table 4: Comparison of defense methods against four backdoor attacks. ACC: clean accuracy; ASR: attack success rate. The first column groups methods by their data requirement. Fixed and oracle denote the threshold selection strategy, as explained in Sec. 5.1. The last column shows average clean accuracy / average attack success rate over the four attacks. Best and second best result per column in **bold** and underlined, respectively.

| Data | Method | BadNet | | Blend | | Refool | | WaNet | | Average |
|------|--------|--------|--------|--------|--------|--------|--------|--------|--------|---------|
| | | ACC | ASR | ACC | ASR | ACC | ASR | ACC | ASR | ACC / ASR |
| – | Backdoored | 69.24 | 99.00 | 68.46 | 99.45 | 68.22 | 96.35 | 69.29 | 98.29 | 68.68 / 97.41 |
| Training | FT | 64.86 | 97.66 | 66.83 | 57.78 | 65.45 | 12.89 | 66.41 | 0.08 | 65.89 / 42.10 |
| | FP | 62.45 | 98.21 | 64.59 | 44.29 | 63.50 | 28.29 | 63.31 | 0.15 | 63.46 / 42.73 |
| | NAD | 60.80 | 45.76 | 59.90 | 42.49 | 60.20 | 1.02 | 61.10 | 0.09 | 60.50 / 22.34 |
| | BNP (fixed) | 67.02 | **0.00** | 41.73 | 68.00 | **66.93** | 1.02 | 68.65 | 98.25 | 60.98 / 41.82 |
| | BNP (oracle) | 68.74 | 0.09 | 41.76 | 63.86 | 0.11 | **0.01** | **69.26** | 98.25 | 44.97 / 40.55 |
| | ANP (fixed) | 59.39 | **0.00** | 60.05 | 0.05 | 61.61 | 10.08 | 60.72 | 0.07 | 60.44 / 2.55 |
| | ANP (oracle) | **69.12** | 0.01 | **67.31** | 0.03 | 61.61 | 10.08 | 68.26 | 0.01 | 66.58 / 2.53 |
| Calibration | OTBR (fixed) | 60.75 | **0.00** | 64.95 | **0.01** | 62.83 | **0.01** | 62.39 | 89.09 | 62.73 / 22.28 |
| | OTBR (oracle) | 60.75 | **0.00** | 64.95 | **0.01** | 62.83 | **0.01** | 62.39 | 89.09 | 62.73 / 22.28 |
| | CLP (fixed) | 36.97 | 0.47 | 26.19 | 9.94 | 37.36 | 6.59 | 40.75 | 0.07 | 35.82 / 4.27 |
| | CLP (oracle) | 10.68 | 0.15 | 7.23 | **0.01** | 59.57 | 2.47 | 56.38 | 0.12 | 33.47 / 0.69 |
| | C&C (oracle) | 54.06 | 96.98 | 57.25 | 98.21 | 57.88 | 90.27 | 60.25 | 98.57 | 57.36 / 96.01 |
| Data-free | CMP (Ours) | 66.90 | 0.01 | 66.70 | 0.05 | 66.67 | 0.16 | 66.58 | **0.00** | **66.71** / 0.06 |

we choose four distinct attacks, BadNets (Gu et al., 2019), Blend (Chen et al., 2017), Refool (Liu et al., 2020), and WaNet (Nguyen & Tran, 2021). Note that, as mentioned in Sec. 2, more recent dynamic attacks that require generators do not converge on the full ImageNet, so are excluded. All attacks are conducted on either the full ImageNet-1K (Russakovsky et al., 2015) or the 10-class subset ImageNet-10, using ResNet-18 (He et al., 2016). The poisoning rate is set to $10\%$ for all attacks except WaNet, for which we use a reduced rate of $2\%$ to prevent significant degradation in performance on clean data. The target label is set to the first class. All models are trained using the SGD optimizer (Robbins & Monro, 1951) with a learning rate of 0.1, momentum of 0.9, and a batch size of 256, following a multi-step learning rate schedule over 90 epochs.

**Defense Settings** We compare CMP agains eight existing defense methods, which are fine-tuning, fine-pruning (Liu et al., 2018a), NAD (Li et al., 2021a), ANP (Wu & Wang, 2021), BNP (Zheng et al., 2022b), OTBR (Lin et al., 2025), CLP (Zheng et al., 2022a), and C&C (Phan et al., 2024b). For methods that use the training set or the poisoned samples, we assume access to $1\%$ of the training data. OTBR, CLP and C&C are data-free methods, but it still require a calibration set to select the optimal pruning threshold. The selected baselines cover all range of fine-tuning, pruning, and unlearning based methods, and those that require the training data and data-free methods. All codes are based on BackdoorBench (Wu et al., 2025). For CMP, we synthesize 1000 images per class using model inversion (Yin et al., 2020; Lee et al., 2024) and train a student network from scratch with the AdamW optimizer (Kingma & Ba, 2015; Loshchilov & Hutter, 2019). We also employ CMP as an OOD-based defense algorithm, where we leverage COCO (Lin et al., 2014) as the OOD dataset. We test our defense on ResNet-18 (He et al., 2016) and ViT (Touvron et al., 2021; Liu et al., 2021) architectures.

**Evaluation Metrics** We evaluate each defense method using two metrics: clean accuracy on the validation (clean) set and the attack success rate (ASR) on triggered images. ASR is defined as the percentage of non-target-class samples that are misclassified into the target class after applying the trigger. For fair comparison, we report the clean accuracy corresponding to an ASR of $0.1\%$ (i.e., $1/C$) whenever possible. If this target ASR is not achievable, we instead report the best available results via cherry-picking.

**Threshold Selection** Instead of using a single, fixed threshold for competing methods, we evaluate CMP against baselines that have been optimized with two favorable threshold selection strategies. (1) A unified cherry-picked threshold shared across all attack types, and (2) a per-attack oracle threshold.

For the first strategy, we select the threshold that achieves the best overall performance while maintaining an ASR reasonably close to the $1/C$ criterion. For the second strategy, we directly choose the threshold that yields an ASR closest to $1/C$ for each attack. *Both strategies represent an overly favorable and unrealistic setting*, as it assumes oracle knowledge of incoming attacks. As shown in Fig. 2, existing defenses exhibit severe sensitivity to threshold choice, which undermines their robustness in practical deployment. For both strategies, we perform a sweep over 20 threshold values; details are provided in appendix.

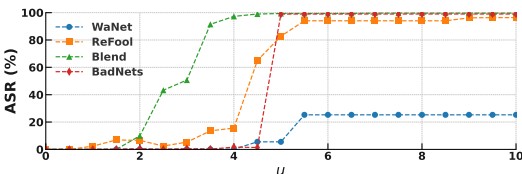

Figure 2: Threshold sensitivity of CLP (Zheng et al., 2022a) against various types of attack methods. We plot the ASR respect to different values of chosen $u$.

### 5.2 PERFORMANCE EVALUATION

**Main Results** We compare CMP against state-of-the-art defense methods on ResNet-18 (He et al., 2016) attacked on ImageNet-1K (Russakovsky et al., 2015) in Table 4. The results show that CMP achieves competitive performance across all evaluated attack types and metrics and state-of-the-art performance on average, without requiring clean data or threshold tuning. CMP reduces the attack success rate (ASR) to approximately $0.1\%$ or lower for Badnets, Blend, Refool, and Wanet, while maintaining benign accuracy around $2\%$ of the original poisoned model. Although baseline methods such as BNP or OTBR can sometimes match or slightly outperform CMP on specific attacks when a carefully tuned threshold is used, their performance varies significantly depending on the incoming attack. The thresholds required for optimal defense differ for each attack, and no single threshold generalizes across scenarios. Also, the pruning-based methods are inconsistent, and sometimes fails to mitigate backdoors such as WaNet completely. As a result, these methods are not robust to unknown or varying attack types, whereas CMP maintains consistently strong performance without relying on threshold selection or hyperparameter finetuning.

These results validate three key observations. First, CMP maintains robust performance across diverse attacks without requiring any external data or threshold tuning. Second, no method other than CMP is capable of suppressing ASR below the $1/C$ criterion, even with a per-attack oracle threshold. Third, CMP consistently achieves better or comparable clean accuracy to existing defenses, establishing it as the most practical and effective solution for fully data-free backdoor removal.

**Small Classes and All-to-All Attacks** To further assess the robustness of CMP, we validate on ImageNet-10 and all-to-all (A2A) attacks. Note that A2A attacks do not converge on the full ImageNet-1K due to the shear number of classes. Table 5 shows that CMP outperforms other defenses even on small datasets and A2A attacks. This strengthens our claim that CMP is a robust and practical defense, whose effectiveness is not dependent on specific dataset characteristics or attack patterns.

Table 5: Performance of CMP against A2O and A2A in small class scenario. 'o' denotes oracle threshold.

| Method | BadNet (ATO) | | BadNet (ATA) | | Avg. |
|---|---|---|---|---|---|
| | ACC | ASR | ACC | ASR | ACC/ASR |
| Backdoored | 92.10 | 98.90 | 98.90 | 90.60 | 95.5/94.8 |
| OTBR (o) | 54.20 | 2.00 | 83.20 | 5.20 | 68.7/3.6 |
| BNP (o) | **92.00** | 1.60 | **92.20** | **1.40** | **92.1**/1.5 |
| CLP (o) | 16.60 | 8.90 | 18.60 | 6.00 | 17.6/7.5 |
| ANP (o) | 91.20 | 2.90 | 91.60 | 2.00 | 91.4/2.5 |
| CMP (ours) | 91.60 | **0.30** | 91.40 | 2.40 | 91.5/**1.4** |

**Cleansing Clean Models** Next, we evaluate the effect of applying CMP to a clean model in Table 6. We generate synthetic images from a clean ResNet-18 network, and follow the same training procedure as in Sec. 4. We find that the resulting student model maintains clean accuracy within two percent of the original model and does not introduce any significant degradation. This result demonstrates that our method does not adversely affect clean models and can be used as a safe sanity-check procedure even when the underlying model is not poisoned.

Table 6: Applying CMP to a clean model preserves clean accuracy within 2%.

| Model | ACC (Before) | ACC (After CMP) |
|---|---|---|
| Clean ResNet-18 | 69.2% | 67.4% |

**Ablation Studies** We report ablation studies of CMP in Table 7 on the backdoor attack Refool (Liu et al., 2020). The results demonstrate that both our mismatch and skew filtering lower the final ASR, and combining them drops the ASR the most. Notably, incorporating filtering not only improves backdoor suppression but also enhances overall performance when used together with skew filtering. This indicates that filtering not only mitigates backdoor behaviors but also guides the selection of semantically meaningful inputs. Finally, we find that detaching the second-highest logit during distillation introduces minimal

Table 7: Ablation study of mismatch and skew filtering on Refool Liu et al. (2020). Both components reduce ASR, and their combination achieves the lowest ASR with minimal impact on clean accuracy.

| Mismatch Filtering | Skew Filtering | ACC (%) | ASR (%) |
|:---:|:---:|:---:|:---:|
| ✗ | ✗ | 67.34 | 91.05 |
| ✓ | ✗ | 66.75 | 85.27 |
| ✗ | ✓ | 65.89 | 0.26 |
| ✓ | ✓ | 66.67 | **0.16** |

performance degradation, but effectively suppresses backdoor behavior.

**CMP to OOD defense and transformers** Since CMP's training procedure is decoupled from the data preparation stage, it naturally extends to an out-of-distribution (OOD) defense setting. In particular, CMP can utilize any publicly available OOD images to train a student model through our knowledge-distillation pipeline. This enables a more practical deployment, especially for architectures without batch normalization layers, such as transformers (Vaswani et al., 2017; Liu et al., 2021; Brown et al., 2020). While previous OOD knowledge-distillation defense failed without model pruning (Pang et al., 2023), we successfully adapt the mismatch filter to control the Bayesian prior $p(y)$ for each label. Specifically, we adopt uni-

Table 8: Performance of CMP using OOD data (COCO Lin et al. (2014)) against BadNet Gu et al. (2019) and WaNet Nguyen & Tran (2021). ResNet-18 He et al. (2016) is employed as the poisoned teacher.

| Role | Model | BadNet | | WaNet | |
|:---|:---|:---:|:---:|:---:|:---:|
| | | ACC (%) | ASR (%) | ACC (%) | ASR (%) |
| Teacher | ResNet-18 | 69.24 | 99.00 | 69.29 | 98.29 |
| Student | ResNet-18 | 58.73 | 0.02 | 57.88 | 0.00 |
| | DeiT-Tiny | 46.17 | 0.00 | 45.83 | 0.00 |
| | Swin-Tiny | 64.36 | 0.00 | 63.21 | 0.00 |

form distribution as the target marginal distribution, and track top-10 predicted classes in each mini-batch and prevent any single class from dominating.

We report the performance of OOD-based CMP in Table 8. To evaluate the performance, we use representative methods BadNets (Gu et al., 2019) for static triggers and WaNet (Nguyen & Tran, 2021) for dynamic triggers, executed with ResNet-18 as the poisoned teacher. This is because the existing ViT training recipe is computationally expensive, and in practice, it is hard to find a recipe that poisons a model well and also attains benign accuracy. Therefore, we only consider distillation scenario. We assume the poisoned teacher is ResNet-18, and use both ResNet-18 (same architecture), DeiT-Tiny (Touvron et al., 2021) and Swin-Tiny (Liu et al., 2021) (ViT variants) as students. Using only the COCO (Lin et al., 2014) dataset as OOD input, CMP reduces the ASR nearly 0 for all settings. These results demonstrate 1) our method is architecture-agnostic, and 2) our method is effective even in distillation settings where existing methods fail. Note that although we naively use COCO directly, which leads to some performance degradation, considering that current state-of-the-art KD methods can recover performance through augmentation from just a single image (Asano & Saeed, 2021), this suggests that we can extremely reduce ASR while fully recovering clean performance.

## 6 CONCLUSION

In this paper, we present CMP, the first universal backdoor defense that is fully data-free, threshold-free, and scalable to practical, large-scale deployments. Unlike prior methods that rely on unrealistic assumptions such as access to training data or tuned thresholds, CMP requires nothing but the poisoned model itself. It consistently removes backdoors across a wide range of attacks on ImageNet while maintaining high clean accuracy. Because CMP decouples data generation from training, it can operate with out-of-distribution data such as COCO, enabling generalization to architectures without batch normalization, including transformers like DeiT. These results highlight CMP as a practical and robust solution for real-world vendor scenarios.

**Reproducibility Statement** In the Appendix, we provide detailed configurations and training settings. Our experimental setup closely follows the guidelines established in prior works on backdoor attacks, so reproducing these baselines directly translates to reproducing our method without additional difficulty. We provide detailed algorithm in Appendix for reproducibility.

**Societal Impact Statement** This research enhances AI security by providing a practical defense against backdoor attacks without requiring calibration data, making AI systems more trustworthy for deployment in critical applications like autonomous driving, medical diagnostics, and security systems. By eliminating the need for threshold tuning, CMP democratizes access to effective backdoor defenses, benefiting organizations with limited data availability. However, like many security-focused research efforts, our techniques could potentially be adapted to remove intentional watermarks or other benign "backdoor-like" features that might be used for copyright protection or model attribution.

**Usage of LLMs** This paper utilized LLMs sparsely, primarily for grammatical error correction and text polishing. All content was verified by the authors, who bear full responsibility for the final manuscript.

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

---

**Algorithm 1** Calibration-Free Model Purification (CMP)

---

1: **Input:** Poisoned model $f_\theta^p$
2: **Output:** Purified student model $f_\phi^s$
3: **Notation:** $X$: synthetic dataset, $\tilde{X} \subset X$: sampled batch, $\mathcal{B}$: augmentation, $\mathbf{s}^p$: teacher's soft label, $\mathcal{F}_{\mathrm{mm}}$: mismatch filter, $\mathcal{F}_{\mathrm{sk}}$: skew filter
4: Generate synthetic dataset $X$ using Eq. (4)
5: **for** each training epoch **do**
6:     Sample mini-batch $\tilde{X} \subset X$
7:     Apply augmentations: $\mathcal{B}(\tilde{X}) = \{a_i(\tilde{x}_i) \mid \tilde{x}_i \in \tilde{X}\}$
8:     Compute soft labels: $\mathbf{s}^p(\mathcal{B}(\tilde{x})) = \mathrm{softmax}(f_\theta^p(\mathcal{B}(\tilde{x})))$
9:     Apply mismatch filtering $\mathcal{F}_{\mathrm{mm}}$ as in Eq. (5) and skew filtering $\mathcal{F}_{\mathrm{sk}}$ as in Eq. (6)
10:     Update $f_\phi^s$ by minimizing the total loss in Eq. (7)
11: **end for**
12: **return** $f_\phi^s$

---

## A  THRESHOLD SELECTION

We provide full details of the performance of ANP Wu & Wang (2021) and CLP Zheng et al. (2022a) with respect to the chosen thresholds. We sweep over 20 values for both methods, and report the clean accuracy (ACC) and attack success rate (ASR) for the four baseline attacks in Fig. 3 and Fig. 4. We employ two strategies as reported in Table 2, where we choose a fixed threshold for all attacks, or a per-attack oracle threshold. For the fixed threshold, we select a value that achieves best ACC with ASR reasonably close to $1/C$, and for the oracle threshold, we choose values that achieves ASR closest to $1/C$.

## B  IMPLEMENTATION DETAILS

| config | value |
|---|---|
| optimizer | SGD |
| base learning rate | 0.1 |
| momentum | 0.9 |
| weight decay | 1e-4 |
| batch size | 256 |
| lr step size | 30 |
| lr gamma | 0.1 |
| training epoch | 90 |
| augmentation | RandomResizedCrop |

| config | value |
|---|---|
| optimizer | AdamW |
| base learning rate | 1e-3 |
| weight decay | 1e-2 |
| batch size | 128 |
| learning rate schedule | cosine decay |
| training epoch | 300 |
| augmentation | RandomResizedCrop |

Table 9: Training configurations for ResNet (left) and KD (right).

**Attack settings**   We train a ResNet-18 He et al. (2016) with the official training recipe as in Table 9 (Left). For backdoor attacks badnets Gu et al. (2019), blend Chen et al. (2017), and refool Liu et al. (2020), we poison $10\%$ of the dataset, and $2\%$ for wanet Nguyen & Tran (2021). This is because a high poison rate would degrade the clean accuracy substantially. For badnets Gu et al. (2019), we use a $20 \times 20$ resolution white square placed on the right-bottom corner. We use a $0.2$ mixing ratio for blend Chen et al. (2017).

**Defense settings**   We follow model inversion Yin et al. (2020) and knowledge distillation Hinton et al. (2015) settings commonly used in the knowledge distillation based dataset distillation works Yin et al. (2023); Yin & Shen (2024); Sun et al. (2024). Note that *we do not tune these hyperparameters, and simply use them off-the-shelf* to demonstrate that our method is robust and does not require extensive hyperparameter tuning. The full details are provided in Table 9 (Right).

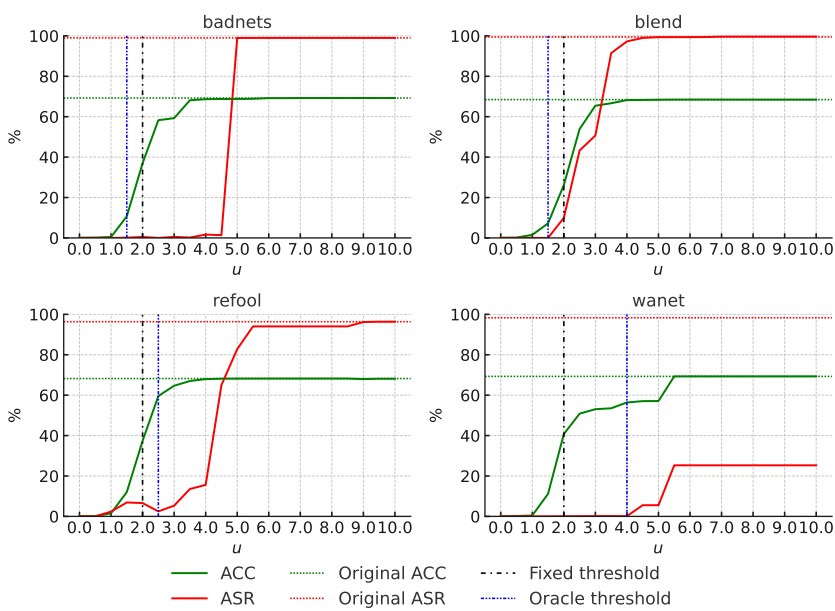

Figure 3: Performance of CLP Zheng et al. (2022a) against four baseline attacks. We plot ACC and ASR with respect to different threshold values, along with the chosen fixed and oracle threshold.

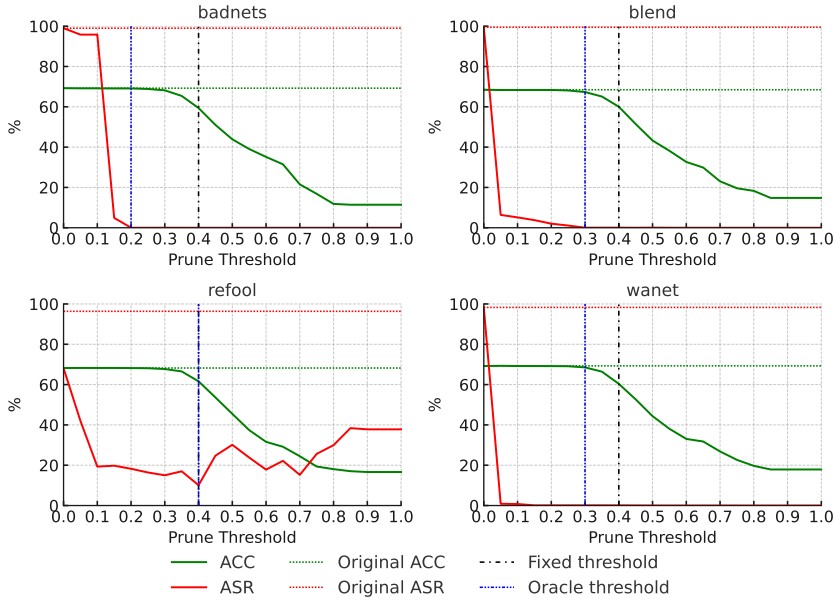

Figure 4: Performance of ANP Wu & Wang (2021) against four baseline attacks. We plot ACC and ASR with respect to different threshold values, along with the chosen fixed and oracle threshold.

# C    TYPES OF BACKDOOR ATTACKS

In this section, we provide a comprehensive overview of backdoor attack taxonomies and justify our experimental setup that uses four representative attack methods: BadNets Gu et al. (2019), Blend Chen et al. (2017), Refool Liu et al. (2020), and WaNet Nguyen & Tran (2021). These methods were carefully selected to cover the full spectrum of backdoor attack types, ensuring that our evaluation of the CMP defense method is thorough and generalizable.

## C.1    ATTACK TAXONOMY

Backdoor attacks can be categorized along several dimensions, with the most significant being the trigger type and its relationship to the input data. Based on these characteristics, we classify backdoor attacks into two primary categories: static and dynamic attacks.

## C.2    STATIC ATTACKS

Static attacks use triggers that remain consistent across all poisoned inputs. These can be further subdivided based on the trigger's spatial characteristics:

**Local Trigger Attacks    Representative method: BadNets Gu et al. (2019)**

Local trigger attacks embed a small, spatially concentrated pattern (such as a patch or symbol) at a fixed position in the input image. Key characteristics include:

- Fixed position, typically in a corner or less salient region of the image
- Consistent pattern across all poisoned samples
- Relatively easy to implement but also easier to detect
- Direct modification of pixel values in a localized region

BadNets exemplifies this approach by inserting a small, fixed pattern (e.g., a square pattern) at a consistent location. This attack is highly effective but leaves a distinctive signature that more sophisticated defenses can identify.

**Global Trigger Attacks    Representative method: Blend Chen et al. (2017)**

Global trigger attacks distribute the trigger pattern across the entire input, affecting all or most of the image area. Key characteristics include:

- Pattern spans the entire input space rather than a localized region
- Often implemented through blending or transparency effects
- More subtle visual appearance compared to local triggers
- May be less disruptive to semantic content

The Blend attack achieves this by superimposing a pattern across the entire image with controlled transparency, making it less visually obvious than patch-based attacks while maintaining high attack success rates.

**Multiple Trigger Attacks    Representative method: Refool Liu et al. (2020)**

Multiple trigger attacks employ several different triggers that all map to the same target label. Key characteristics include:

- Uses multiple distinct trigger patterns
- All triggers direct classification to the same target
- Increases attack robustness against defenses that may only identify some triggers
- Often leverages naturally occurring visual elements

Refool specifically uses reflection patterns as triggers, embedding multiple types of reflection effects that appear natural to human observers but strongly influence model predictions. This makes the attack both stealthy and robust against many defense mechanisms.

## C.3 DYNAMIC ATTACKS

**Representative method: WaNet Nguyen & Tran (2021)**

Unlike static attacks, dynamic attacks use input-dependent triggers that adapt to the specific content of each image. Key characteristics include:

- Trigger transformation depends on the input image content
- No fixed pattern that repeats across poisoned samples
- Often implemented through subtle warping or transformations
- Significantly harder to detect through statistical analysis

WaNet implements this approach through imperceptible image warping, where the warping function creates a trigger that is uniquely applied to each image. This makes the attack particularly challenging to defend against, as traditional methods that search for consistent patterns across poisoned inputs fail to identify the dynamic nature of the trigger.

## C.4 JUSTIFICATION FOR SELECTED ATTACK METHODS

Our evaluation utilizes these four attack methods (BadNets, Blend, Refool, and WaNet) because they:

1. **Span the trigger design space**: From simple local patterns (BadNets) to sophisticated input-adaptive transformations (WaNet)
2. **Represent varying levels of detectability**: From more obvious modifications to nearly imperceptible changes
3. **Exercise different attack mechanisms**: Direct pixel manipulation, global blending, naturalistic artifacts, and geometric transformations
4. **Challenge different aspects of defense systems**: Some attacks may be neutralized by specific defense components while being resistant to others

By evaluating CMP against this diverse set of attack methods, we ensure that our defense is robust against the full spectrum of backdoor attacks, rather than being optimized for a particular attack subtype. This comprehensive evaluation approach provides stronger evidence for the universality of our defense method to both known and potentially unknown backdoor attack variants.

# D  QUALITATIVE RESULTS

In this section, we provide qualitative results of augmentation and model inversion. Specifically, we visualize the cases for which augmentations lead to "direct poisoning" (Trigger-like behavior), and also examples of synthesized images obtained via model-inversion.

## D.1 SYNTHETIC IMAGES

We provide in Fig. 5 and Fig. 6 visualizations of synthetic images obtained via model inversion for BadNet and Refool networks. We can see there no visual distinctions between the two group of images.

## D.2 AUGMENTATIONS

We visualize cases for which clean images (images that are classified to their original class) are misclassified to the target class after augmentations. This provides us with an understanding of how

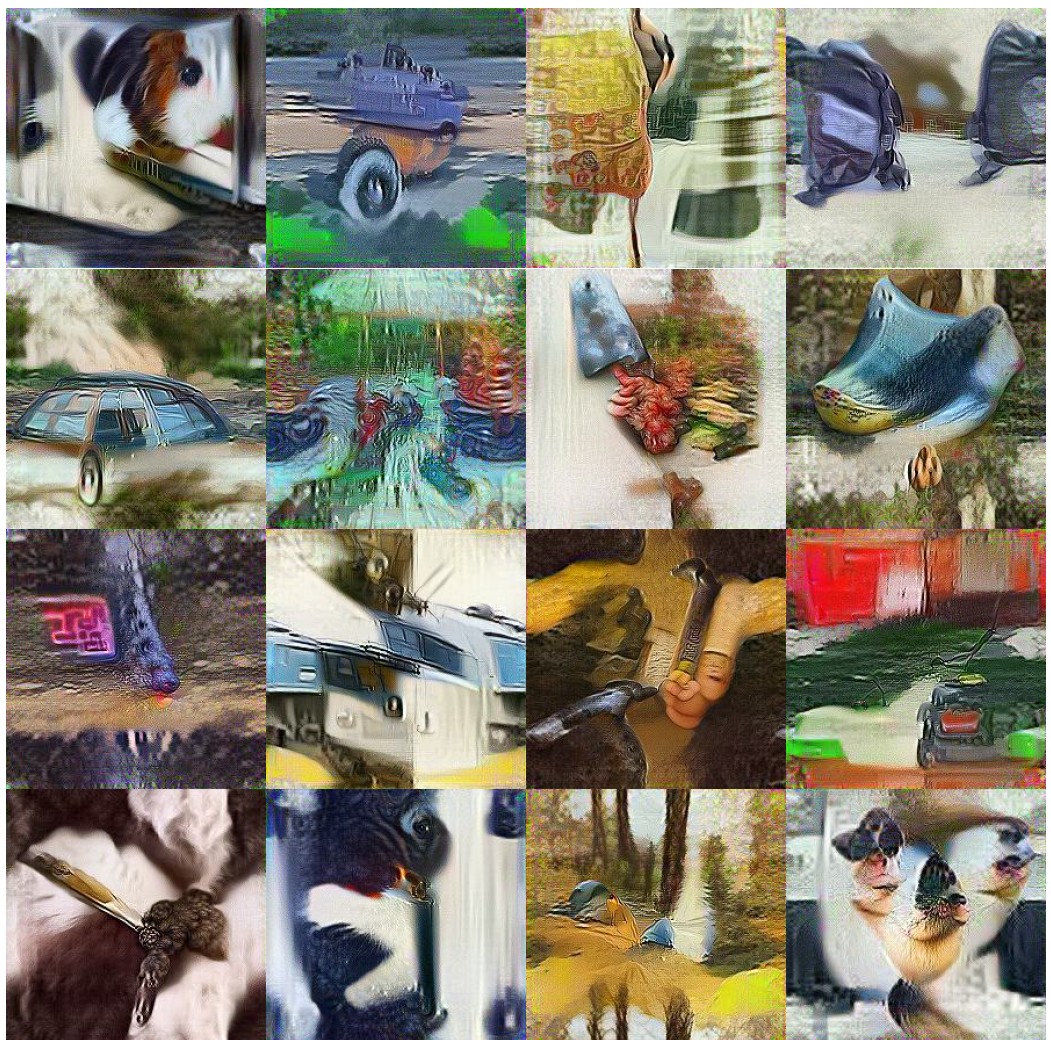

Figure 5: Synthetic images obtained via model inversion on the BadNet network.

standard augmentations can lead to poisoned behaviors. As shown in Fig. 7 and Fig. 8, images that have white box like patches on the bottom right corner become classified as the target class. This is because the patch acts like a trigger when passed through the badnet network. Similarly, images in Fig. 9 also gets mislabeled as the target class. Although we cannot exactly identify which components lead to trigger like images (although we suspect high-frequency components induce it), due to the complexity of the trigger of refool, this shows that seemingly benign samples can trigger poisoned image behaviors, demonstrating the necessity of our mismatch filter. The mismatch filter removes all samples that are mislabled, removing these harmful examples from the training pipeline.

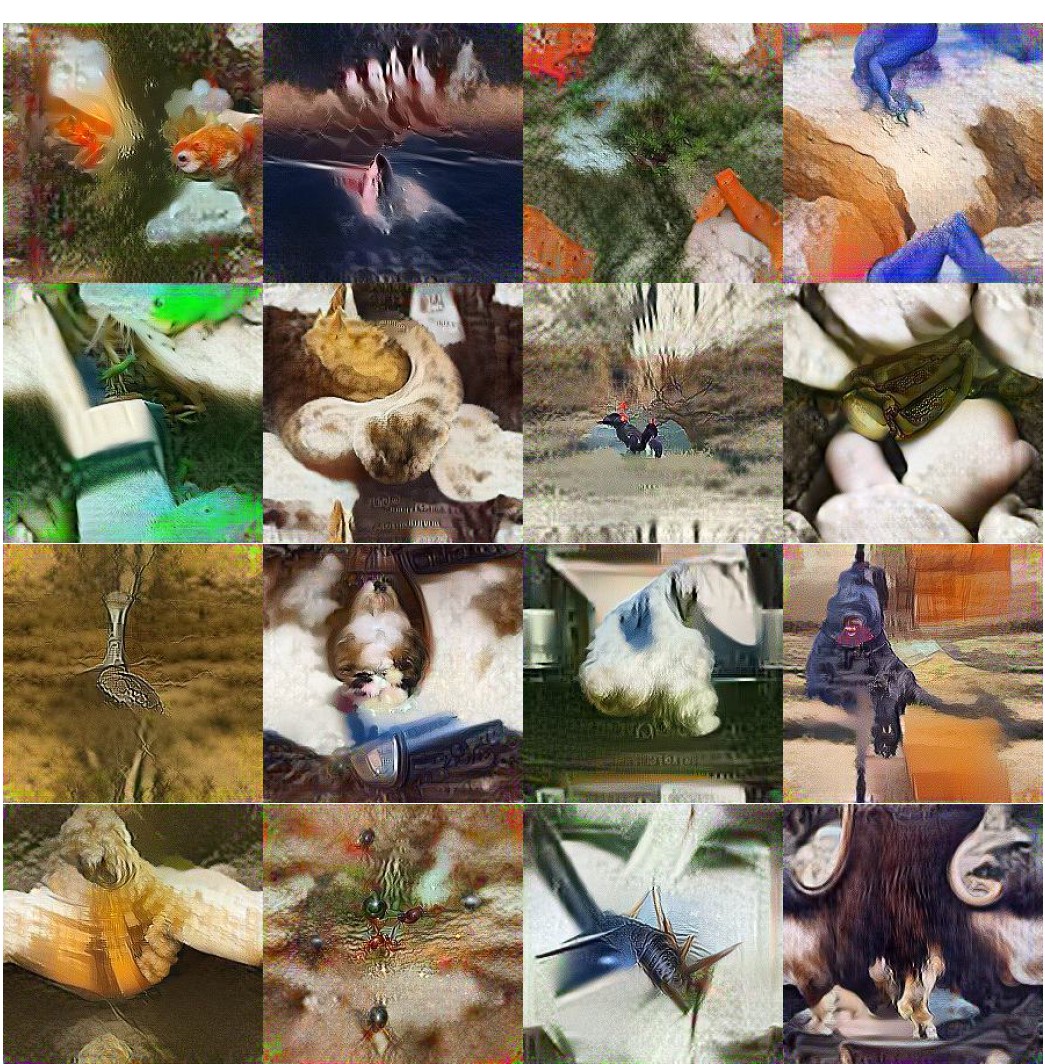

Figure 6: Synthetic images obtained via model inversion on the Refool network.

**Original**                    **RandomSizedReCrop**

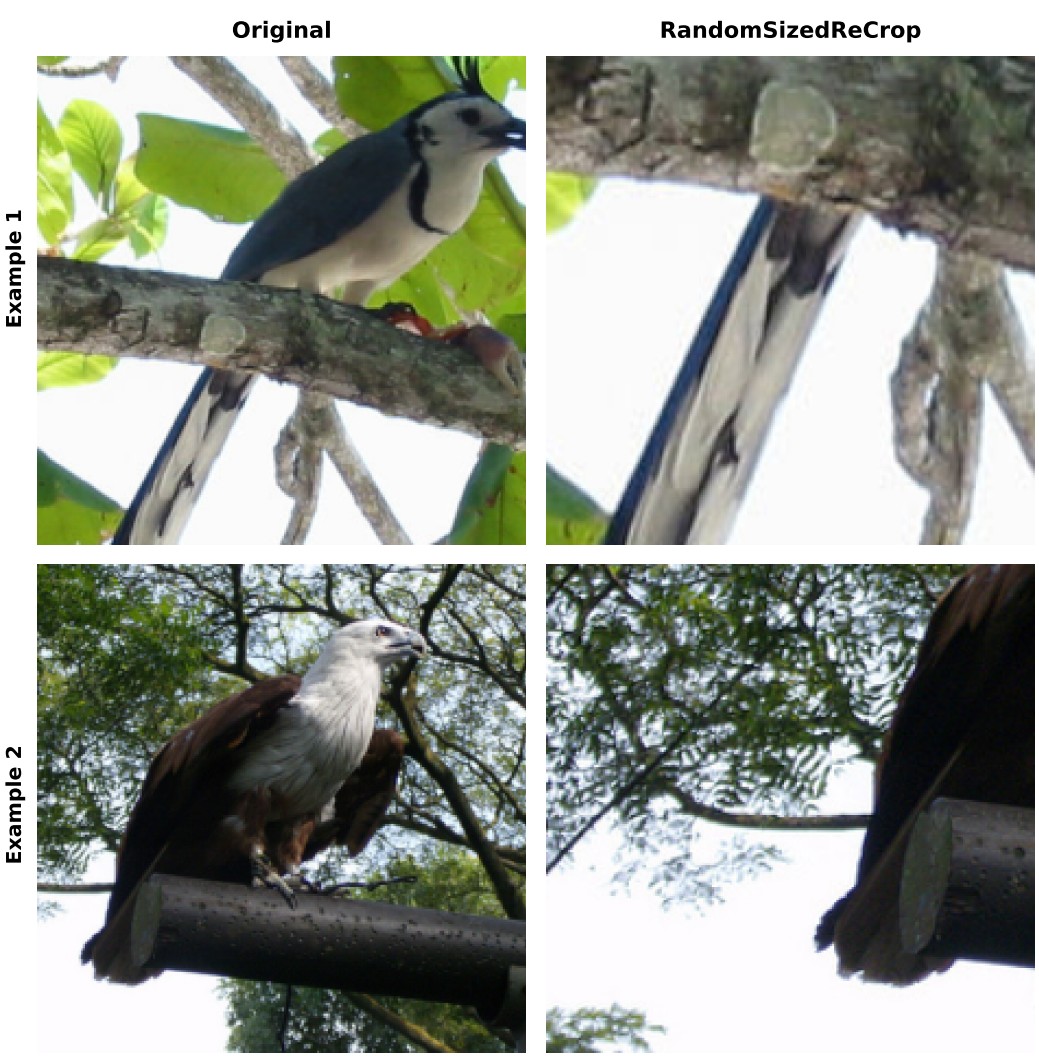

Figure 7: Visualization of clean ImageNet images being misclassified as the target class after the RandomSizedReCrop augmentation on BadNet.

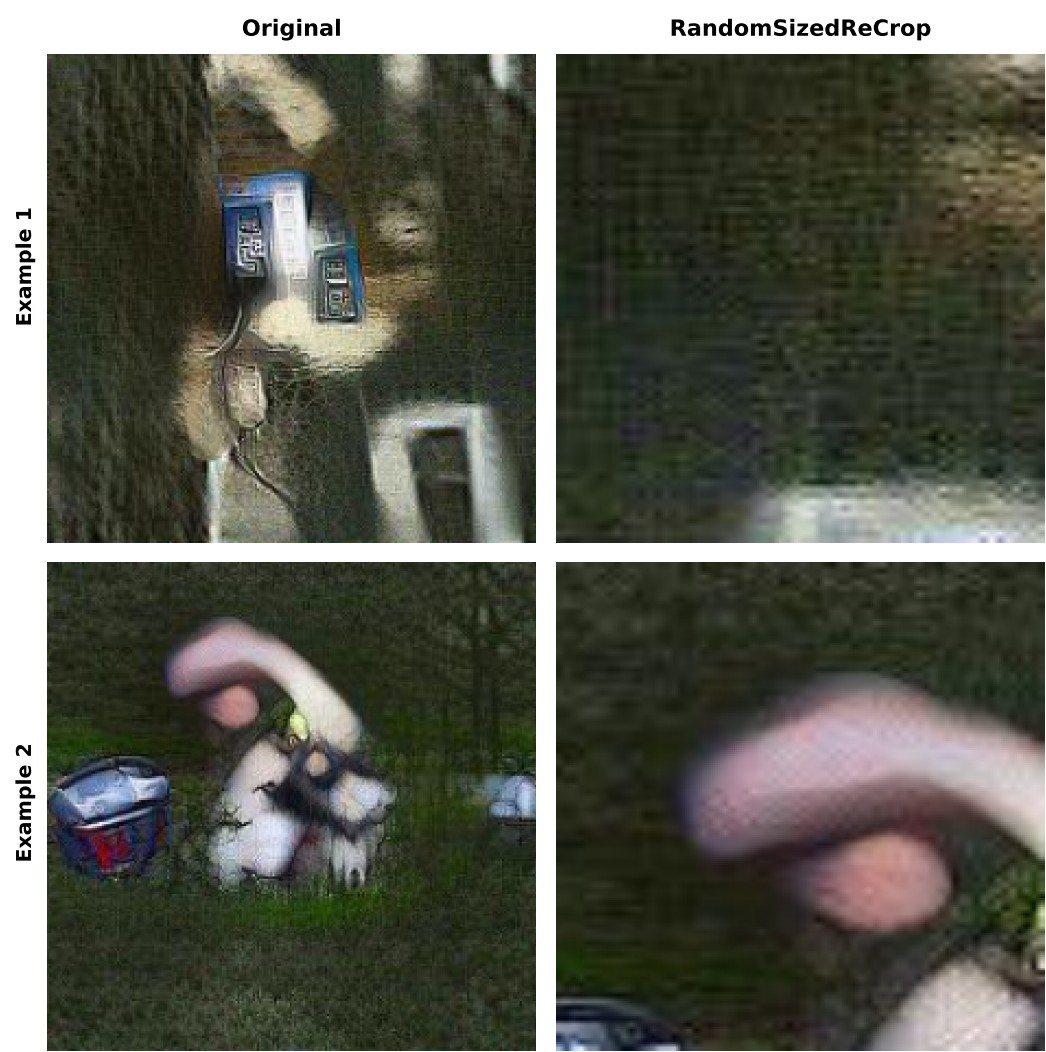

Figure 8: Visualization of synthetic images being misclassified as the target class after the Random-SizedReCrop augmentation on BadNet.

Original                          RandomSizedReCrop

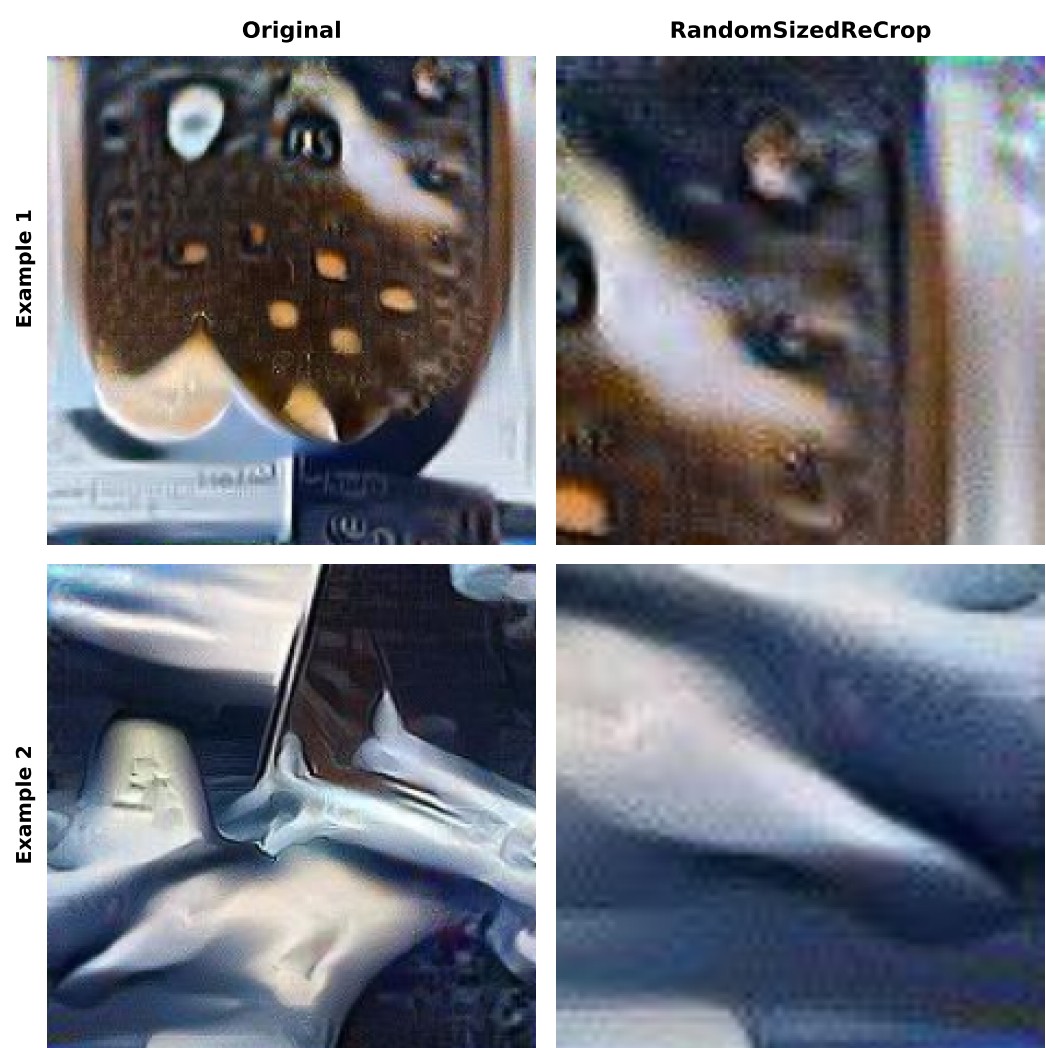

Figure 9: Visualization of synthetic images being misclassified as the target class after the Random-SizedReCrop augmentation on Refool.