# OpenReview forum: "Calibration-Free Defense Against Backdoor Attacks in the Wild"
_ICLR.cc/2026/Conference — Submitted to ICLR 2026_

### Official Review · Reviewer_NQnR · 2025-10-21

**Soundness:** 4
**Presentation:** 3
**Contribution:** 4
**Rating:** 8
**Confidence:** 3

**Summary:**

Exisiting backdoor defense methods rely on pruning with the assumption that backdoors are encoded in a small set of specific neurons. This paper argues against this assumption for tis ineffectiveness on large-scale models and proposes Calibration-free Model Purification. It avoids purning entirely and leverages a self-distillation framework guided by the discovery of a systematic "prediction skew". A dual filtering system is employed. CMP reduces attack success rates on diverse attacks while preserving clean accuracy.

**Strengths:**

- The motivation of the work is very interesting. The authors identify two fundamental limitations from exisiting defenses, including assumption on backdoors encoded in a small subset of identifiable neurons and unavailability due to requring training resources form threshold calibration. Then a calibration-free model purification method is proposed. I think these observations are helpful to this area.
- The proposed CMP method employs two filters. They are reasonable and easy to implement. Across different backdoor attacks, the proposed method shows consistent better performances.
- The authors also show experiments on OOD data. This shows the generalizability of the proposed method.

**Weaknesses:**

- I did not find major weaknesses of this paper. Overall, I think the paper is well-motivated and presented.

**Questions:**

- In Figure 1, should the label for Teacher model in 2. Mismatch Filter be the Target Label?
- In Sec. 3.2, it mentions augmentations cause benign images to exhibit trigger-like characteristics. Which augmentation operators are evaluated? Does this conclusion apply to a set of augmentations or some specific augmetation operators?

---

> ### Author Response · Authors · 2025-11-21
>
> We sincerely thank Reviewer NQnR for the encouraging and positive evaluation. We are grateful that you appreciated the core motivation of our work—addressing the limitations of pruning-based defenses and the unavailability of calibration data. We are also glad that you found our dual-filtering system reasonable and our OOD experiments to be a strong demonstration of generalizability. We have addressed your specific questions regarding Figure 1 and the augmentation operators below.
>
> **[Q1] In Figure 1, should the label for Teacher model in 2. Mismatch Filter be the Target Label?**
>
> We thank the reviewer for this question, which allows us to clarify the diagram. To be precise, the label predicted by the teacher model does not have to be the Target Label. However, as shown in Table 2, the "direct poison rate" (the rate at which misclassified images are incorrectly labeled as the target class) is 17.05% under standard augmentations. This rate is over 1700 times higher than the baseline misclassification rate of a clean teacher (which is ~0.1% for ImageNet's 1000 classes). Therefore, it is highly likely that a sample misclassified by the poisoned teacher is, in fact, a poisoned sample that is falsely labeled as the target class. The Mismatch Filter is designed to catch exactly this scenario and eliminate these harmful samples from the training pipeline.
>
> **[Q2] In Sec. 3.2, it mentions augmentations cause benign images to exhibit trigger-like characteristics. Which augmentation operators are evaluated? Does this conclusion apply to a set of augmentations or some specific augmentation operators?**
>
> We thank the reviewer for this question, which allows us to clarify our experimental setup. For detailed qualitative examples, please refer to the common answer [C3].
> The augmentations evaluated in Section 3.2 (and shown in Table 2) are RandomResizedCrop and CutMix. These are standard augmentations commonly used in self-distillation pipelines.
>
> We found that these  transformations can inadvertently generate patches or spatial arrangements that, while derived from benign images, mimic trigger-like characteristics for the poisoned model. This effect activates the backdoor behavior, leading to the "direct poisoning" effect we report.
>
> This is quantified in Table 2, which shows a dramatic increase in the direct poisoning rate (e.g., from 1.47% to 17.05% for synthetic data) when these augmentations are applied. We have also included several qualitative examples to provide further visual intuition for this phenomenon [C2].

---

### Official Review · Reviewer_ua98 · 2025-10-25

**Soundness:** 2
**Presentation:** 2
**Contribution:** 1
**Rating:** 2
**Confidence:** 4

**Summary:**

This work proposes a data-free backdoor defense method, CMP, which utilizes a self-distillation framework to prevent a student model from learning the malicious behaviors of a teacher model. The results indicate that CMP is effective for backdoor defense. However, the experimental evaluation is incomplete: the main performance evaluation was conducted solely on ImageNet, whereas the analysis for the key motivation in the main text used CIFAR-10. Additionally, the paper lacks a discussion on adaptive attacks.

**Strengths:**

1，	The knowledge distillation process can cause a student model to inherit the backdoor behaviors of its teacher. Addressing this issue, this work proposes a more robust and secure distillation framework.
2，	This work utilizes "Model Inversion" to synthesize data from the model itself, requiring no additional clean datasets.

**Weaknesses:**

1，	The notion that backdoors can transfer from a teacher to a student model via knowledge distillation has been discussed in prior works [1] [2]. However, in the "RELATED WORKS" section, the authors claim to have "uncovered an unprecedented mechanism that allows backdoors to transfer." It is recommended that the authors review and clarify their novelty claim here.
2，	This work finds that data augmentation can cause benign images to exhibit trigger-like characteristics. Although the authors demonstrate this phenomenon in Table 2, they do not provide a more detailed analysis or discussion. It is suggested to add visualizations or other more interpretable analyses.
3，	In its related work on 'Backdoor Defense', this paper discusses pruning-based defenses more extensively. However, the core method of this work is self-distillation. It is recommended that the authors include more summary of similar methods [1-6].
4，	What are the time and computational costs of model inversion, and what is the visualization of the synthetic image samples?
5，	In Section 3.2, the authors design an experiment on the CIFAR-10 dataset to demonstrate the "prediction skew" phenomenon. However, in the experimental evaluation in Section 5, the authors do not provide the defense performance results of their method on CIFAR-10.

**Questions:**

1，	The paper states in Section 5.1 that the CMP method synthesizes 1000 images per class, which implies that for the ImageNet-1K dataset, approximately 1,000,000 synthetic images need to be generated. What is the time cost of this process?
2，	The paper lacks a discussion on adaptive attacks. If an attacker anticipates that the defender will use distillation for purification, they could design a novel backdoor to bypass this defense. Notably, [7] proposed Anti-Distillation Backdoor Attacks (ADBA), which are specifically designed to survive the knowledge distillation process and successfully transfer from the teacher model to the student.

The citation of weakness and questions are listed below.
[1] Yao Z, Zhang H, Guo Y, et al. Reverse backdoor distillation: Towards online backdoor attack detection for deep neural network models[J]. IEEE Transactions on Dependable and Secure Computing, 2024, 21(6): 5098-5111.
[2] Hong J, Zeng Y, Yu S, et al. Revisiting data-free knowledge distillation with poisoned teachers[C]//International Conference on Machine Learning. PMLR, 2023: 13199-13212.
[3] Yoshida K, Fujino T. Disabling backdoor and identifying poison data by using knowledge distillation in backdoor attacks on deep neural networks[C]//Proceedings of the 13th ACM workshop on artificial intelligence and security. 2020: 117-127.
[4] Hu C, Teng X, Xing W, et al. Distill To Detect: Amplifying Anomalies in Backdoor Models through Knowledge Distillation[C]//ICASSP 2025-2025 IEEE International Conference on Acoustics, Speech and Signal Processing (ICASSP). IEEE, 2025: 1-5.
[5] Bie R, Jiang J, Xie H, et al. Mitigating backdoor attacks in pre-trained encoders via self-supervised knowledge distillation[J]. IEEE Transactions on Services Computing, 2024, 17(5): 2613-2625.
[6] Li X, Gao Y, Liu M, et al. A New Data-Free Backdoor Removal Method via Adversarial Self-Knowledge Distillation[J]. IEEE Internet of Things Journal, 2024.
[7] Ge Y, Wang Q, Zheng B, et al. Anti-distillation backdoor attacks: Backdoors can really survive in knowledge distillation[C]//Proceedings of the 29th ACM International Conference on Multimedia. 2021: 826-834.

---

> ### Author Response · Authors · 2025-11-21
>
> We sincerely thank Reviewer ua98 for the critical and comprehensive review. We appreciate your recognition of our robust distillation framework and the value of our model inversion approach. We also acknowledge your valid concerns regarding the novelty of the backdoor transfer phenomenon, the completeness of our related work comparisons, and the consistency of our experimental benchmarks. We have carefully addressed these points below, particularly by clarifying our novelty regarding "prediction skew" and providing the requested cost analysis.
>
> **[W1] The notion that backdoors can transfer from a teacher to a student model via knowledge distillation has been discussed in prior works.**
>
> We apologize for the confusion caused by the phrasing "uncovered an unprecedented mechanism." We realize that this expression may have been interpreted as claiming novelty for the general phenomenon of backdoor transfer itself. We would like to clarify that our use of the term "mechanism" refers specifically to the underlying cause (i.e., the prediction skew) that enables transfer even under clean-data conditions, rather than the transferability phenomenon observed in prior works.
>
> 1. Clarification of Scope against Prior Works ([1] & [2]) We acknowledge the existence of prior works [1] and [2], but their investigations are grounded in premises that fundamentally differ from ours. Our work specifically challenges the conventional assumption that backdoored models are safe to use for distillation if the data is clean (not triggered).
>
> Regarding [1]: This work operates on the straightforward premise that backdoor transfer is a natural consequence of data poisoning. In their setting, the transfer occurs because the dataset contains poisoned samples. Consequently, their contribution focuses on filtering out these poisoned data points. This logic implicitly reinforces the conventional view: "If you remove the poison, you stop the transfer." They do not address the scenario where transfer occurs without any poisoned data, which is the core of our investigation.
>
> Regarding [2]: This work demonstrates transfer via Out-of-Distribution (OOD) data. Crucially, this approach adheres to the basic definition of backdoor attacks, that the model "acts normally" on clean, in-distribution data. By relying on OOD data to trigger the transfer, they implicitly accept the premise that the backdoor remains dormant (and thus does not transfer) when the data is clean and in-distribution (line 210-212).
>
> In contrast, our work challenges this fundamental premise. We demonstrate that the "act normal" assumption is deceptive in the context of knowledge distillation. As shown in our results, even when the student acts on purely clean, in-distribution data, the teacher's "Stealthy Prediction Skew" (our identified mechanism) actively transfers the backdoor. This finding that the backdoor transfers in the exact scenario where it is theoretically supposed to be dormant, is the "unprecedented" aspect we aimed to highlight.
>
> 2. Clarification of the "Unprecedented Mechanism" Our claim of discovering an "unprecedented mechanism" refers specifically to the "Stealthy Prediction Skew" found in clean data knowledge-distillation, not the general observation of transfer itself. We have rewritten the relevant section to highlight this chain of discovery:
>
> 1. Unprecedented Finding with Clean Data: First, we demonstrate an unprecedented phenomenon. As shown in Table 1, the backdoor transfers to the student model (ASR 22.23%) even when distilling using only clean real images. This is a critical finding, as prior work often implies transfer happens due to trigger-like inputs. Our experiment shows the backdoor persists even on a completely clean dataset, which is an unexplained phenomenon that motivates our investigation.
>
> 2. The Mechanism is "Stealthy Prediction Skew": We then identify the mechanism for this. Prior work often implies transfer happens when the student mimics the teacher on trigger-like inputs. We find the mechanism is more subtle. As shown in Section 3.2, poisoned models exhibit a systematic bias where they assign abnormally high probabilities to the target label even for correctly classified, benign inputs . The student model learns this "inflated Bayesian prior," which is the fundamental mechanism for backdoor transfer. This skew explains the (otherwise mysterious) transfer on clean data from our first finding.

---

> ### Author Response · Authors · 2025-11-21
>
> **[W3] The core method of this work is self-distillation. It is recommended that the authors include more summary of similar methods**
>
> We thank the reviewer for this constructive suggestion.
> Our extensive discussion of pruning-based defenses stems from the fact that they represent the primary and most dominant paradigm in data-free backdoor defense , which is the central problem setting of our work. Many recent methods that are "data-free" still rely on pruning, so comparing against them was a priority.
>
> However, the reviewer is correct that a more detailed comparison with other distillation-based methods will strengthen the paper. In the current draft, we do discuss existing knowledge-distillation (KD) attempts for cleansing, such as those [1]. We also include NAD in our experimental comparisons. A key differentiator, which we will elaborate on, is that these prior KD-based methods still rely on a pruning step to be effective and fail without them. Our method, CMP, is the first to propose a pure self-distillation and filtering framework that is effective without any pruning or threshold calibration.
>
> Following the reviewer's advice, we will revise the "Related Works" section to expand this discussion. We will create a more distinct subsection for "Distillation-Based Defenses" to better survey these methods and use this to more clearly contextualize the novelty of our pruning-free and calibration-free approach.
>
> [1] Pang, Lu, et al. "Backdoor cleansing with unlabeled data." Proceedings of the IEEE/CVF Conference on Computer Vision and Pattern Recognition. 2023.
>
> **[W5] The authors do not provide the defense performance results of their method on CIFAR-10.**
>
> We chose CIFAR-10 for the analysis in Section 3.2 primarily for its simplicity and widespread usage as a benchmark to clearly illustrate the "prediction skew" phenomenon. It was not intended to hide performance issues or imply that our method is dataset-specific.
>
> To address the reviewer's concern and demonstrate the universality of this phenomenon, we have expanded our validation to **ImageNet-10**, a more complex subset of ImageNet. The results indicate that the "prediction skew" mechanism is not only present but **even more pronounced** in this challenging setting.
>
> Experimental Validation on ImageNet-10:
> We conducted the "Trigger-Free Attack" (transferring backdoor via skewed soft labels only) on ImageNet-10.
>
> - **Baseline (Clean):** The model achieved a Clean Accuracy (CA) of **76.20%** and an ASR of **2.79%** (below random guessing).
>
> - **With Skewed Soft Labels:** When trained with the skewed soft labels (mimicking the teacher's bias), the model achieved a CA of 68.20% but the **ASR surged to 66.28%**.
>
> This confirms that the backdoor transfer driven by prediction skew is a robust phenomenon, essentially replicating an architecture-agnostic attack. The fact that ASR jumps from ~2.8% to ~66.3% purely due to soft label distribution validates the necessity of our defense mechanism (CMP), which is designed to neutralize this skew.
>
> 3. Causal Validation via Trigger-Free Attack: We are the first to provide causal evidence for this mechanism. In our "trigger-free attack" experiment (Table 3), we trained a model on clean CIFAR-10 images, using only skewed soft labels (e.g., 70% correct, 30% target class) . This model, which never saw a trigger, achieved a ~70% ASR when tested with one. This demonstrates that the prediction skew itself is sufficient to create the backdoor vulnerability, independent of any triggers in the distillation data.
> Therefore, our novelty lies in this complete reasoning: (1) observing the unprecedented transfer on clean real data, (2) identifying the "prediction skew" as the mechanism, and (3) proving its causality. Our defense, CMP, is consequently novel because its filters are designed not just to catch triggers, but to explicitly neutralize this "prediction skew."
>
> Action Taken: To address this and avoid ambiguity, we have rewritten the Novelty paragraph in the Related Works (and Introduction) section. We explicitly replaced the vague term "unprecedented mechanism" with a precise description of our findings regarding clean data transfer and prediction skew.
>
> Original Text:
> "(2) Novelty: It differs from existing works that rely on pruning backdoored neurons, as we uncover an unprecedented mechanism that allows backdoors to transfer. We mitigate this problem using a self-distillation framework and a two-step filtering mechanism."
> Revised Text:
>
> "(2) Novelty: It differs from existing works that rely on the presence of poisoned or OOD data. We first uncover that backdoors can be transferred even on a clean, in-distribution dataset, going a step beyond prior findings that imply transfer is restricted to OOD contexts. We identify 'Stealthy Prediction Skew' as the underlying mechanism driving this phenomenon. We mitigate this problem using..."

---

> ### Author Response · Authors · 2025-11-21
>
> **[W2] This work finds that data augmentation can cause benign images to exhibit trigger-like characteristics. It is suggested to add visualizations or other more interpretable analyses.**
>
> We thank the reviewer for this valuable suggestion. The reviewer is correct that this is a critical phenomenon, and a purely quantitative analysis, as presented in Table 2, is best supported by qualitative, interpretable evidence. To provide this exact analysis, we have included several qualitative visualizations in the common answer [C3].
>
> These visualizations illustrate how standard augmentations (such as RandomResizedCrop) can inadvertently crop patches from a benign image in a way that the poisoned model misinterprets as a trigger-like feature. This visual evidence provides the intuition behind the quantitative spike in the "direct poisoning" rate shown in Table 2. Together, they demonstrate why our Mismatch Filter is a necessary component. It is not only filtering for poisoned images, but for these benign-yet-transformed images that activate the backdoor behavior. We added a sentence in Section 3.2 to explicitly refer the reader to these visualizations in the Appendix.
>
> Action Taken:
> We have included these ImageNet-10 validation results in Section 3.2
>
> **[W4 and Q1] The paper states in Section 5.1 that the CMP method synthesizes 1000 images per class. What is the time cost of this process?**
>
> We thank the reviewer for raising this practical concern regarding the computational resources required for our method. Please refer to the common answer [C4] for a detailed breakdown of the computational costs.
>
> **[Q2] If an attacker anticipates that the defender will use distillation for purification, they could design a novel backdoor to bypass this defense.**
>
> We thank the reviewer for pointing out this important oversight. We agree that discussing adaptive attacks like ADBA [7] is crucial for demonstrating the robustness of our defense.
> To address this, we have implemented ADBA on the CIFAR-10 dataset and evaluated our defense (CMP) against it. The results and our detailed analysis are as follows:
>
> 1. Analysis of ADBA: We reproduced the ADBA attack. The resulting student model achieved a Clean Accuracy (CA) of 88.79% and an Attack Success Rate (ASR) of 93.83% without defense. During our analysis, we observed that the ADBA optimization process—which iteratively updates a single trigger to maximize transferability—essentially converges towards a Targeted Universal Trigger, similar to architecture-agnostic attacks like TAP. This implies that ADBA actually generates Targeted Universal Adversarial Perturbation, not a backdoor attack.  To verify this, we tested the generated trigger on a standard CIFAR-10 model (not distilled via ADBA). Surprisingly, it yielded a 94.98% CA and 89.10% ASR, confirming that ADBA functions by generating a robust, highly transferable universal perturbation.
>
> 2. Defense Performance of CMP Despite the sophisticated nature of ADBA, we applied our proposed CMP defense (introducing our two-step filters) under the exact same KD settings. The results again verify that CMP can be applied universally.
>
> Without Defense: ASR 93.83% / CA 88.79%
> With CMP: ASR 7.59% / CA 70.11%
>
> 3. Mechanism of Defense: As discussed in our Global Response, CMP works by explicitly neutralizing the "Stealthy Prediction Skew." Even though ADBA is designed to survive standard distillation, it relies on embedding strong directional features that create high confidence (skew) towards the target class. By filtering out these skewed predictions and altering the marginal distribution of the student's learning targets, CMP drastically increases the information quantity required for the trigger to be effective. Consequently, the adaptive trigger fails to transfer, reducing the ASR from ~94% to ~7.6%.
>
> Action Taken: We have added a new subsection "Robustness against Adaptive Attacks" in the Discussion section. Here, we include the ADBA experiment results and the analysis explaining how CMP breaks the ADBA mechanism by neutralizing prediction skew.

---

### Official Review · Reviewer_LcoR · 2025-10-25

**Soundness:** 2
**Presentation:** 3
**Contribution:** 3
**Rating:** 2
**Confidence:** 3

**Summary:**

This paper proposes CMP (Calibration-Free Model Purification) — a data-free, threshold-free defense against neural backdoor attacks. The key insight is that backdoor persistence during knowledge distillation arises not merely from specific neurons but from a systemic prediction skew, where the poisoned teacher model over-assigns probability mass to the target class even on benign inputs. CMP adopts a self-distillation framework using synthetic images generated via model inversion.

**Strengths:**

- The paper is well-organized and clearly written.
- The paper introduces a new causal explanation of backdoor transfer through prediction skew, shifting the perspective from neuron-level pruning to distribution-level bias correction.
- The dual-filter design is conceptually novel and operationally elegant, effectively removing backdoors without any calibration data.

**Weaknesses:**

- The “prediction skew” phenomenon, though empirically compelling, lacks a formal definition or an information-theoretic grounding. No bound is provided linking skew magnitude to ASR reduction. A KL-based or mutual information analysis would strengthen the theoretical foundation.
- Experiments are restricted to moderate-size architectures (ResNet-18, small ViT/DeiT). The paper claims scalability to larger models but does not report training cost, memory, or wall-time for full ImageNet distillation. This is critical for real-world viability.
- The attacker is assumed fixed; CMP’s robustness against adaptive threats is not explored.

**Questions:**

- Why is detaching the second-highest logit the most effective choice?
- How sensitive is CMP to the quality or diversity of DeepInversion-generated samples? Does the defense remain stable if inversion produces low-quality or partially mode-collapsed data?
- Please report distillation runtime, GPU hours, and resource overhead relative to standard fine-pruning or NAD.

**Details Of Ethics Concerns:**

N / A

---

> ### Author Response · Authors · 2025-11-21
>
> We sincerely thank Reviewer LcoR for the constructive criticism and for recognizing the conceptual novelty and operational elegance of our dual-filter design. We value your insight that our work shifts the perspective from neuron-level pruning to distribution-level bias correction. We acknowledge your valid concerns regarding the theoretical formalization of "prediction skew" and the computational scalability of the method. We have carefully addressed these points below, including a new formal analysis and a detailed cost breakdown.
>
> **[W1 and Q1] The "prediction skew" phenomenon lacks a formal definition or an information-theoretic grounding.**
>
> We appreciate the reviewer for highlighting the need for a formal definition and theoretical grounding to strengthen the paper's foundation. Regarding the mathematical justification and formal definition of the prediction skew, please refer to our common answer [C1, C2].
>
> **[W2 and Q3] The paper claims scalability to larger models but does not report training cost, memory, or wall-time for full ImageNet distillation.**
>
> We thank the reviewer for raising this critical point regarding the real-world viability of our method. For a detailed breakdown of the training costs, memory usage, and wall-time for ImageNet distillation, please refer to our common answer [C4].
>
> **[W3] The attacker is assumed fixed; CMP’s robustness against adaptive threats is not explored.**
>
> Thank you for your suggestion and concern on CMP's robustness. As stated in Section C in the Appendix, we carefully divided existing backdoor attacks into four parts, then selected their representatives in the supplementary material. The four attacks cover the entire spectrum of attack designs. Also, we included adaptive attacks such as WaNet, a highly sophisticated and difficult-to-detect dynamic attack that uses imperceptible image distortions and varies its trigger based on input data. More importantly, we note that we have attempted to conduct extensive experiments on other adaptive attacks such as IAB and SSBA. Both methods require training a generator for triggering inputs. However, even with extensive hyperparameter tuning, we could not get the training to converge as either the loss for the generator exploded to infinity or completely disrupted the downstream training process. We suspect this is due to the limitations of the attack itself, this is because input-aware backdoor methods are essentially same as implementing generator which creates adversarial noise with given images. However, due to curse of dimensionality, image scale noise generators are not implemented yet, and recent works still relies on adaptive optimization [1] or universal perturbations [2], which aligns with static trigger (blended, badnet)
>
> [1] Salman et al, Raising the Cost of Malicious AI-Powered Image Editing (ICML 2023)
>
> [2] Ahn et al, Nearly Zero-Cost Protection Against Mimicry by Personalized Diffusion Models (CVPR 2025)
>
> **[Q2] How sensitive is CMP to the quality or diversity of DeepInversion-generated samples?**
>
> We thank the reviewer for this insightful question regarding the robustness of our method. For qualitative results of the synthetic images, please refer to the common answer [C3].
>
> 1 On Sensitivity to Synthetic Samples' Quality: CMP is not fundamentally coupled to a specific high-quality inversion methods. In our experiments, we intentionally employed SRe2L, the most naive approach in the field of self-distillation [4]. The field of self-distillation is well-researched, and our choice demonstrates that CMP is effective even without relying on a SOTA inversion methods. We hypothesize that employing more advanced, SOTA distillation techniques would likely produce a stronger, more diverse set of samples, which would further improve CMP's performance rather than degrade it [5, 6, 7]. Our current results, achieved with the simplest baseline, demonstrates a strong robustness against the synthetic images.
>
> 2. On Stability Against Low-Quality or Mode-Collapsed Data: CMP's mismatch filter is designed to handle specifically this scenario. If the inversion process fails (producing low-quality or "posioned" samples), our proposed OOD detection module identifies and filters out these samples. This filtering stage acts as a safeguard, ensuring that only representative, high-fidelity data is used. Also, as mentioned above, SOTA methods for model-inversion and self-distillation have various methods to prevent mode-collapse. We highlight that even if the model inversion fails, the core of CMP's algorithm is in the filering process. If all model inversion methods fails, due to excessive amount of low-quality samples or mode collapse, CMP can be extended seamlessly to the OOD method, as shown in Section 5. As OOD images can be taken from any source, this provides a robust fallback that is entirely independent of the inversion-based generator's quality or stability.

---

> ### Author Response · Authors · 2025-11-21
>
> [4] Yin, Zeyuan, Eric Xing, and Zhiqiang Shen. "Squeeze, recover and relabel: Dataset condensation at imagenet scale from a new perspective." Advances in Neural Information Processing Systems 36 (2023): 73582-73603.
>
> [5] Shao, Shitong, et al. "Elucidating the design space of dataset condensation." Advances in Neural Information Processing Systems 37 (2024): 99161-99201.
>
> [6] Du, Jiawei, et al. "Diversity-driven synthesis: Enhancing dataset distillation through directed weight adjustment." Advances in neural information processing systems 37 (2024): 119443-119465.
>
> [7] Cui, Jiacheng, et al. "FADRM: Fast and Accurate Data Residual Matching for Dataset Distillation." arXiv preprint arXiv:2506.24125 (2025).

---

### Official Review · Reviewer_g5od · 2025-10-28

**Soundness:** 3
**Presentation:** 3
**Contribution:** 3
**Rating:** 6
**Confidence:** 4

**Summary:**

The paper proposes Calibration-Free Model Purification (CMP), a fully data-free and threshold-free defense mechanism against backdoor attacks in deep neural networks. CMP leverages a self-distillation framework with two novel components — the mismatch filter and skew filter — to prevent the transfer of backdoor behaviors from a poisoned teacher to a clean student model. It operates without requiring calibration data or pruning, achieving robust results on ImageNet-scale experiments across diverse attacks. The authors claim CMP is the first practical, scalable, and data-free defense adaptable to both CNNs and transformers.

**Strengths:**

- The idea of removing the backdoor without calibration data is novel and practically important. The paper contributes new insights into the mechanisms of backdoor transfer during knowledge distillation.
- The method is well-motivated with analysis experiments in section 3.2, which also makes the presentation better for understanding.
- The extensive experiments verified the effectiveness of CMP. Its architecture-agnostic design strengthens its broader impact.

**Weaknesses:**

- While the prediction skew is identified as the key phenomenon, the paper offers limited formal or mathematical justification for why the dual-filtering scheme guarantees elimination of such skew under different attack types. Especially for the skew filter, which relies heavily on the assumption of the second-largest probability class as the backdoor class, it lacks empirical evidence.
- Despite the synthesized data is from the previous technique (i.e., model inversion), it lacks visualization or explanation of how different they look compared to the real image. And more experiments are needed to validate the effectiveness of CMP using different model inversion techniques, illustrating the influence of the synthesized data.
- The required LLM statement is not included in both the main text and appendix.

**Questions:**

- The appendix is submitted separately, making it hard to find. As the author guide (https://iclr.cc/Conferences/2026/AuthorGuide) encourages submitting a single file (paper + supplementary text), can you concatenate them?
-  Why do the OTBR (fixed) and (oracle) version performs the same in Table 4? Does it indicate that the calibration data is not important for an effective OTBR?

---

> ### Author Response · Authors · 2025-11-21
>
> We sincerely thank Reviewer g5od for the thoughtful review and constructive feedback. We are encouraged by your appreciation of CMP's novelty in achieving a calibration-free defense and its practical importance for real-world AI security. We are also glad you found our motivation and extensive experiments to be strong contributions. We have carefully addressed your concerns regarding the formal justification of the prediction skew and the qualitative analysis of synthetic data. Please find our detailed responses below.
>
> **[W1] paper offers limited formal or mathematical justification for why the dual-filtering scheme guarantees elimination**
> We thank the reviewer for highlighting the need for a more rigorous formalization of our method. Regarding the mathematical justification, please refer to our common answer [C1, C2] where we have added the requested formal analysis.
>
> **[W2] more experiments are needed to validate the effectiveness of CMP using different model inversion techniques, illustrating the influence of the synthesized data.**
>
> We thank the reviewer for these two constructive suggestions, which will help improve the paper's clarity and completeness.
>
> 1. On Visualization of Synthetic vs. Real Images:
> We agree visualization of synthesized images help to understand our intuition better. For the qualitative examples, please refer to the common [C3] answer. We have uploaded new visualizations in the Appendix.
> 2. On Validating with Different Model Inversion (MI) Techniques:
> Our paper's central claim is that CMP is not fundamentally coupled to a specific high-quality model inversion method.
>  Choice of MI Technique: In our experiments, we adopted SRe2L, a foundational method for self-distillation, rather than a SOTA technique. Our choice demonstrates that CMP is effective even without relying on a SOTA method for data synthesis. We hypothesize that employing more advanced MI or distillation techniques would likely produce a stronger, more diverse set of samples, which would further improve CMP's performance rather than degrade it.
> Strongest Evidence (OOD Experiment): The strongest validation of this claim is already present in our OOD-based defense experiments . In this experiment (Table 8), we completely remove the model inversion component and replace it with a non-generative, external OOD dataset (COCO).
> Results: The results in Table 8 show that this OOD-based approach, which uses data with entirely different characteristics, still reduces the Attack Success Rate (ASR) to near-zero (0.00% - 0.02%) across all student architectures, including Transformers.
> This experiment, which represents a more extreme data-source variation than simply swapping one MI technique for another validates our claim. CMP is robust to the quality and source of the data it uses for distillation. The core of CMP's algorithm is in the filtering process and provides a fallback that is independent of the data synthesis technique's quality or stability.
>
> **[W3] The required LLM statement is not included in both the main text and appendix.**
>
> We are sorry that we have not included the required LLM statement. For the writing of the paper, we have used LLMs only sparsely, mainly for the grammar revision. We have included a full statement about the usage of LLMs in the appendix.
>
> **[Q1] The appendix is submitted separately, making it hard to find.**
>
> We are deeply sorry that we have caused the inconvenience. We have concatenated the appendix to the main pdf. Please check the revised pdf.

---

> ### Author Response · Authors · 2025-11-21
>
> **[Q2] Why do the OTBR (fixed) and (oracle) version performs the same in Table 4?**
>
> We thank the reviewer for this very careful observation. The reviewer is correct that the "OTBR (fixed)" and "OTBR (oracle)" rows in Table 4 are identical.
> This is not an error, but rather a direct result of the threshold selection process. Oracle refers model-wise best threshold, fixed refers best single threshold across models. In this case, modelwise best threshold become same at chance.1. Explanation of "Fixed" vs. "Oracle":
>
>   *  As defined in our paper, the fixed threshold is the single, unified value (cherry-picked from our sensitivity analysis) that provides the best overall average performance across all four attacks.
>
>    *  The oracle threshold is the per-attack value that achieves the best ASR for each specific attack.
>
> 2. Analysis of the Sensitivity Table: The table provided in the query (our threshold sensitivity analysis for OTBR, which we will include in the Appendix) shows why these two selection strategies resulted in the same outcome.
>
>     *  The "fixed" threshold, chosen for the best overall performance, was Gamma = 0.01. This gives the results shown in the "OTBR (fixed)" row.
>     *  When selecting the "oracle" (best-per-attack) threshold, we find:
>         *  For BadNet, Blend, and Refool, Gamma = 0.01 clearly provides the best (or near-best) ASR without destroying accuracy.
>         *  For WaNet, no threshold is effective. As soon as Gamma is increased (e.g., to 0.05), the accuracy plummets from 62.39% to 20.1%. Therefore, the only "viable" (though still failing) oracle choice for WaNet is also Gamma = 0.01 to preserve accuracy.
>
> 3. Because this single threshold (Gamma = 0.01) happened to be both the best overall (fixed) and the best per-attack(oracle) choice for all four attacks in our sweep, the rows in Table 4 are identical.
>
> 4. Is Calibration Data Unimportant for OTBR? To the reviewer's second point: No, on the contrary, this finding demonstrates that calibration data is critically important and a point of failure for OTBR.  Our sensitivity table shows that OTBR is extremely "brittle." Its performance relies entirely on finding one specific, "lucky" threshold (Gamma = 0.01) on a calibration set. Any other choice (e.g., Gamma = 0.05) would lead to a catastrophic performance drop (e.g., BadNet ACC from 60.75% to 24.17%).  This extreme sensitivity to a calibration set—which a real-world user would not have—is the exact vulnerability that our calibration-free method, CMP, is designed to solve.
>
> | Gamma | Badnet   | Blend    | Refool      | Wanet       |
> |----------|----------|----------|-------------|-------------|
> | 0.01     | 60.75/0  | 64.95/0.01 | 62.83/0.01 | 62.39/89.09 |
> | 0.05     | 24.17/0  | 34.79/0  | 53.89/0.02  | 20.1/9.21   |
> | 0.1      | 7.39/0   | 6.88/0   | 38.57/0.24  | 4.45/1.81   |
> | 0.15     | 1.68/0   | 2.49/0   | 19.06/5.62  | 2.13/1.53   |
> | 0.2      | 1.46/0   | 1.1/0    | 1.28/4.74   | 1.11/2.46   |
> | 0.25     | 0.87/0   | 0.37/0   | 0.59/3.18   | 0.57/0.41   |
> | 0.3      | 0.24/0   | 0.34/0   | 0.32/24.58  | 0.59/0.17   |

---

### Author Response · Authors · 2025-11-21
**On the theoretical justification of the probability prediction skew**

**[C1] On the theoretical justification of the probability prediction skew**

### Response to Reviewer regarding the intuition of Prediction Skew

We appreciate the reviewer's suggestion to provide an intuitive explanation for the **prediction skew** phenomenon. We address this concern from an information-theoretic perspective, specifically utilizing **Rate-Distortion Theory**.

1. Theoretical Premise: Simplicity Bias (Occam's Razor)

We assume that neural networks, during training (e.g., via SGD), implicitly minimize the description length of the learned solution (Minimum Description Length principle). In Rate-Distortion terms, the model seeks to minimize the "Rate" (information stored in weights) required to achieve a low "Distortion" (successful attack).

2. The Cost of Learning Subtle Triggers

In advanced backdoor attacks, adversaries aim to embed triggers $\tau$ that are subtle (e.g., invisible patterns or dynamic warping).

- **High Complexity:** Distinguishing these subtle triggers from benign noise requires defining a highly complex decision boundary.

- **Information Cost:** Let $I(X; T)$ be the mutual information required for the model to map the triggered input to the target label $y_t$. For a subtle trigger with a weak signal, $I(X; T)$ is significantly large because the "bit cost" to distinguish the trigger is high.


3. Prediction Skew as an Optimization Shortcut

To minimize the total information cost while maintaining the attack success rate, the model exploits the prediction skew. We can decompose the total information required as follows:

$$\text{Total Information} \approx R_{\text{prior}} + I(X; T \mid \text{skewed prior})$$

- **$R_{\text{prior}} \approx 0$ (Low Cost):** Encoding a global bias (skew) towards the target class $y_t$ is computationally "cheap." It essentially requires adjusting a single bias parameter in the final layer, meaning the information cost $R_{\text{prior}}$ is negligible.

- **$I(X; T \mid \text{skewed prior}) \ll I(X; T)$ (Reduced Conditional Cost):** Once the prior probability for the target class is artificially inflated (skewed), the model requires significantly less evidence from the input $X$ to classify it as $y_t$. The decision boundary lowers, allowing even weak/subtle triggers to cross the threshold easily.


4. Conclusion

Consequently, the model naturally converges to a state exhibiting prediction skew ($\pi_{y_t}^{poisoned} \gg \pi_{y_t}^{clean}$), as this represents the most information-efficient path (Global Minimum in complexity) to satisfy the backdoor objective.

---

### Author Response · Authors · 2025-11-21
**Response regarding the Mechanism of Skew Filter**

**[C2] Response regarding the Mechanism of Skew Filter**

1. Our Strategy: Dual Defense Mechanism

The Skew Filter serves a dual purpose: it simultaneously eliminates the learned prior bias ($R_{\text{prior}}$) and blocks the direct injection of backdoor features.

- **Restoring the Information Barrier:** By removing the skewed prior, we force the information cost required to trigger the backdoor, $I(X; Y_{backdoor})$, back to its original high state.

- **Blocking the Strongest Signal:** It physically blocks the channel where the most potent backdoor signal resides. This effectively creates a "barrier," preventing the student from learning subtle backdoors during distillation.


2. Why the Second-Highest Logit? (Signal Strength Analysis)

The rationale for targeting the second-highest logit stems from the necessity of signal strength for a successful attack.

- **Exclusion of Top-1:** First, the **Mismatch Filter** ensures that no samples explicitly misclassified as the target class enter the distillation process1. Thus, the remaining backdoor signal is **not** in the Top-1 prediction (which corresponds to the correct ground-truth label).

- **Concentration of Malicious Signal:** For a backdoor trigger to be effective, its signal must be strong enough to potentially override the semantic class information. In a correctly classified sample (where the trigger is present but "stealthy"), this high-magnitude signal cannot vanish; it must reside in the **strongest non-target logit**.

- **Conclusion:** Therefore, the second-highest logit is the only candidate capable of carrying a signal strong enough to function as a backdoor. Targeting the Top-1 and Top-2 logits addresses the vast majority of the threat surface.


3. The Role of Detach: Implicit Regularization

Our design choice employs detach (stopping gradients) rather than hard masking (zeroing out). This acts as a form of implicit regularization leveraging the zero-sum dynamics of the Softmax function:

- **Blocking the Inducement:** Consider the competition between the Top-1 (clean) and Top-2 (target) classes. The poisoned teacher sends two simultaneous signals: (A) "Maximize the correct class (Top-1)" and (B) "Maintain high probability for the target class (Top-2)." By detaching the second-highest logit, we block the gradient flow for signal (B)2.

- **Implicit Suppression:** Consequently, the student is no longer encouraged to inflate the second-highest class. As the optimizer maximizes the probability of the Top-1 class (driven by clean features), the probability of the second-highest class is **implicitly suppressed** because the "force" (teacher's skew) holding it up has been removed.

- **Preserving Semantic Integrity (Why not zero out?):** Forcing the logit to zero implies "this class is impossible," which destroys valid semantic relationships (**Dark Knowledge**) between classes (e.g., a dog resembling a wolf). Detach avoids this destruction.


4. Why Only the Second-Highest? (Minimal Intervention)

Our goal is to cleanse the backdoor while maximizing the preservation of clean accuracy. This presents a trade-off between filtering poison-candidates and retaining information quantity.

- **Diminishing Returns:** Extending the filter to the third-highest or lower logits might remove marginal amounts of residual skew poisoning, but it would also discard valid "dark knowledge" essential for student generalization.

- **Optimal Trade-off:** Targeting only the second-highest logit is the **minimal effective intervention**. It removes the logit with the highest probability of being a backdoor carrier while preserving the rich semantic information contained in the rest of the distribution.

---

### Author Response · Authors · 2025-11-21
**On the qualitative results of model inversion based synthetic images, and how augmentations incur backdoor behavior**

**[C3] On the qualitative results of model inversion based synthetic images, and how augmentations incur backdoor behavior**

To address the reviewers' requests for visualizations  and deeper analysis of how augmentations induce backdoor behavior , we have included additional visualizations in the Appendix.

**1. Qualitative Analysis of Synthetic Images**
We provide a visualization of the synthetic images generated via Model Inversion. These images are generated by optimizing the input to match the batch normalization statistics stored in the teacher model. As shown in the figure, these images effectively capture the feature statistics required for distillation, despite containing high-frequency patterns typical of DeepInversion methods.

**2. Mechanism of Augmentation-Induced Backdoor Activation**
We visualize the phenomenon where standard data augmentations inadvertently activate backdoor behaviors, as analyzed in Table 2.
* **Observation:** We specifically visualize the effect of RandomResizedCrop. The figure demonstrates that cropping specific regions of a benign synthetic image can result in a pattern that causes the poisoned teacher to output a high response for the target class (mimicking trigger features).
* **Consequence:** If these augmented samples are blindly used for distillation, the student model learns to associate these benign features with the target label, transferring the backdoor.
* **Justification for Mismatch Filter:** This qualitative analysis empirically justifies our **Mismatch Filter**. By removing samples where the teacher's prediction shifts after augmentation, we effectively filter out these accidental trigger activations.

---

### Author Response · Authors · 2025-11-21
**On the computation costs of generating synthetic images and knowledge distillation**

**[C4] On the computation costs of generating synthetic images and knowledge distillation**

To address concerns regarding the computational overhead of our method, we provide a detailed breakdown of the costs associated with synthetic image generation and knowledge distillation, demonstrating the scalability of CMP.

**1. Cost of Model Inversion (Synthetic Data Generation)**
While generating synthetic data via model inversion might initially appear computationally expensive, the process is highly efficient due to its parallel nature.
* **Parallelization:** The generation process for each class is independent, allowing for "embarrassingly parallel" execution.
* **Hardware Efficiency:** Unlike model training, the inversion process does not require heavy GPU resources and can be effectively parallelized across CPUs. In our experiments, generating the full synthetic dataset (1,000 images per class) takes approximately **18 hours using 10 CPUs**.
* **Contribution Recap:** As stated in line 198-200, our contribution does not rely on the inversion itself, rather, our contribution is to propose generalized As demonstrated in Section 4.2 and Table 8 of the main text, CMP is flexible and can utilize Out-of-Distribution (OOD) data (e.g., COCO) instead of synthetic images. This completely eliminates the generation cost while maintaining robust defense performance.

**2. Cost of Knowledge Distillation**
The distillation phase is comparable to standard model training but can be significantly optimized.
* **Training Time:** For the full ImageNet-1K setting using 1,000 synthetic images per class, our purification process takes approximately **3 days using 4 NVIDIA RTX 3090 GPUs**.
* **Scalability via IPC Reduction:** We further demonstrate that CMP does not require a large number of synthetic images to be effective. We evaluated the performance with reduced Images Per Class (IPC) counts (50, 100, 200, 500). As shown in the table below, we achieve comparable defense performance (low ASR) and clean accuracy even with significantly reduced data, drastically cutting down the computational time.

**Table: Performance of CMP across different Images Per Class (IPC) settings**

| IPC | Clean ACC (%) | ASR (%) | Est. Training Time |
| :--- | :---: | :---: | :---: |
| **50** | *48.9* | *0.11* | **~3.5 Hours** |
| **100** | *53.6* | *0.11* | **~7 Hours** |
| **200** | *56.3* | *0.09* | **~14.5 Hours** |
| **500** | *65.0* | *0.08* | **~36 Hours** |
| **1000 (Default)** | 66.71 | 0.06 | **~72 Hours** |

---

### Author Response · Authors · 2025-12-02
**Summary of Paper Direction, Results & Rebuttal Updates**

**To the Area Chair:**

To assist your assessment, we provide this summary outlining our **core contributions with reviewer consensus**, followed by how we **addressed all major concerns** during the discussion period.

---

### 1. Core Contributions & Reviewer Consensus

**All four reviewers acknowledged the core contributions of our work**, despite differences in final ratings.

**Problem Identification.** We identify two fundamental limitations in existing defenses. 1. The flawed assumption that backdoors reside in pruneable neurons, and 2, the impractical requirement for calibration data. **[Acknowledged: R-NQnR, R-LcoR, R-g5od]**

**Key Discovery.** We discovered "Stealthy Prediction Skew" where poisoned models assign abnormally high probability to the target class even on correctly classified inputs, shifting the perspective from neuron-level pruning to distribution-level bias correction. This prediction skew allows backdoor to transfer during knowledge distillation even when using **clean data.** **[Acknowledged: R-g5od, R-LcoR]**

**Proposed Solution.** We propose CMP, a data-free, threshold-free defense achieving SOTA on ImageNet via a dual-filtering system. Reviewers noted the design as "novel and practically important" [R-g5od], "conceptually novel and operationally elegant" [R-LcoR], and "reasonable and easy to implement" [R-NQnR]. **[Acknowledged: All four reviewers]**

**Experimental Validation.** CMP demonstrates consistent effectiveness across diverse attacks and architectures, with OOD experiments validating generalizability. **[Acknowledged: R-g5od, R-NQnR]**

---

### 2. Addressed Concerns

We conducted extensive new experiments and theoretical analysis to address all major concerns.

**A. Adaptive Attacks [R-ua98, R-LcoR].**
We implemented Anti-Distillation Backdoor Attacks (ADBA) on CIFAR-10. While ADBA achieved 93.83% ASR on undefended models, **CMP reduced ASR to 7.59%** while maintaining clean accuracy.

**B. Theoretical Grounding [R-LcoR, R-g5od].**
We provided formal justification based on the **Minimum Description Length (MDL)** principle: encoding a global bias (skew) is information-efficient compared to encoding complex trigger boundaries. CMP raises this information cost, forcing backdoor removal.

**C. Skew Filter Intuition [R-g5od].**
We clarified why detaching (not zeroing) the second-highest logit is optimal: it blocks the strongest backdoor signal while preserving dark knowledge.

**D. Computational Cost [R-ua98, R-LcoR].**
Generation takes ~18 hours (embarrassingly parallel, reducible to ~10 min with [1] methods). Distillation with **200 IPC achieves 0.09% ASR in 14.4 hours**. Our OOD-based variant eliminates generation costs entirely.

[1] Up to 100x Faster Data-free Knowledge Distillation  (AAAI 2022)

**E. Augmentation Visualization [R-g5od, R-NQnR].**
We provided qualitative examples showing how RandomResizedCrop creates trigger-like patches, validating the necessity of our Mismatch Filter.

---

> ### Author Response · Authors · 2025-12-02
>
> **F. Comparison with Related Works [R-ua98]**
>
> We distinguish CMP from suggested works based on methodology, constraints, and scale. Notably, **all cited methods [1-6] were evaluated only on small-scale datasets (MNIST and CIFAR-10)**, whereas CMP is the first work in this domain to be validated on the full-scale **ImageNet-1K**.
>
> * **[1] Yao et al.** operates on the straightforward premise that backdoor transfer is a natural consequence of data poisoning, thus focusing on **filtering out poisoned data points**. This logic implicitly reinforces the conventional view that removing poison stops transfer. They do not address the scenario where transfer occurs **without any poisoned data**, which is the core of our investigation.
> * **[2] Hong et al.** demonstrates transfer via **Out-of-Distribution (OOD)** data. By relying on OOD data to trigger the transfer, they implicitly accept the premise that the backdoor remains dormant on clean, in-distribution data. in contrast, our work addresses the transfer that occurs even on **clean, in-distribution data**.
> * **[3] Yoshida et al.** requires filtering the training data to remove poisoned samples. However, even if the data is successfully filtered, we demonstrate in **Table 1** that backdoor behaviors still transfer during knowledge distillation **even when using clean training data**. Thus, filtering strategies are insufficient against the distillation-based transfer we address.
> * **[4] Hu et al.** does not use knowledge distillation for **cleansing** a model. Instead, it posits that student models trained via KD with poisoned teachers become backdoored, and then leverages the discrepancy between the student and teacher to **detect** poisoned samples. CMP, conversely, is a model sanitization method.
> * **[5] Bie et al.** utilizes knowledge distillation but violates the data-free constraint by **requiring access to the training data**. Furthermore, the scope of [5] is limited to cleansing **pretrained encoders**, whereas CMP is designed to defend end-to-end classifier models.
> * **[6] Li et al.** requires training **two GAN-style networks** (one for generating clean samples and one for poisoned samples). GANs are notoriously difficult to train, and training two simultaneously makes the method highly unreliable. Additionally, it introduces significant hyperparameter complexity, requiring the selection of **5 different thresholds**, whereas CMP is threshold-free.
>
>
>
>
> | Method | Clean-Data Backdoor Transfer | Data-Free | Threshold-Free | ImageNet Scale | Model Purification |
> |:-------|:----------------------------:|:---------:|:--------------:|:--------------:|:------------------:|
> | [1] Yao et al. | ✗ | ✗ | ✗ | ✗ | ✓ |
> | [2] Hong et al. | ✗ | ✓ | ✓ | ✗ | ✓ |
> | [3] Yoshida et al. | ✗ | ✗ | ✗ | ✗ | ✓ |
> | [4] Hu et al. | ✗ | ✓ | ✗ | ✗ | ✗ (Detection) |
> | [5] Bie et al. | ✗ | ✗ | ✗ | ✗ | ✗ (Encoder) |
> | [6] Li et al. | ✗ | ✓ | ✗ | ✗ | ✓ |
> | **CMP (Ours)** | **✓** | **✓** | **✓** | **✓** | **✓** |
>
>
>
>
> [1] Yao Z, Zhang H, Guo Y, et al. Reverse backdoor distillation: Towards online backdoor attack detection for deep neural network models[J]. IEEE Transactions on Dependable and Secure Computing, 2024, 21(6): 5098-5111. [2] Hong J, Zeng Y, Yu S, et al. Revisiting data-free knowledge distillation with poisoned teachers[C]//International Conference on Machine Learning. PMLR, 2023: 13199-13212. [3] Yoshida K, Fujino T. Disabling backdoor and identifying poison data by using knowledge distillation in backdoor attacks on deep neural networks[C]//Proceedings of the 13th ACM workshop on artificial intelligence and security. 2020: 117-127. [4] Hu C, Teng X, Xing W, et al. Distill To Detect: Amplifying Anomalies in Backdoor Models through Knowledge Distillation[C]//ICASSP 2025-2025 IEEE International Conference on Acoustics, Speech and Signal Processing (ICASSP). IEEE, 2025: 1-5. [5] Bie R, Jiang J, Xie H, et al. Mitigating backdoor attacks in pre-trained encoders via self-supervised knowledge distillation[J]. IEEE Transactions on Services Computing, 2024, 17(5): 2613-2625. [6] Li X, Gao Y, Liu M, et al. A New Data-Free Backdoor Removal Method via Adversarial Self-Knowledge Distillation[J]. IEEE Internet of Things Journal, 2024.
>
> ### 3. Summary
>
> | Concern | Raised By | Addressed In | Key Result |
> |---------|-----------|--------------|------------|
> | Adaptive attacks | R-ua98, R-LcoR | Rebuttal [A] | ADBA ASR: 93.8% → **7.6%** |
> | Theoretical grounding | R-LcoR, R-g5od | Rebuttal [B] | MDL-based formalization |
> | Skew filter intuition | R-g5od | Rebuttal [C] | Detach vs. zero-out analysis |
> | Computational cost | R-ua98, R-LcoR | Rebuttal [D] | **14.4h** with 200 IPC |
> | Augmentation visualization | R-g5od, R-NQnR | Rebuttal [E] | Qualitative examples added |

---

### Meta-Review · Area_Chair_aECF · 2025-12-13

**Summary:**

Reviewers generally agreed that the paper tackles an important and practical problem—calibration-free, data-free backdoor defense at realistic scale—and found the proposed CMP framework technically sound and empirically strong. However, several concerns influenced the final assessment.

The primary concern was theoretical rigor and clarity. While the notion of prediction skew as the mechanism enabling backdoor transfer during distillation was considered insightful, some reviewers felt the original paper lacked a sufficiently formal definition and theoretical justification. The rebuttal added information-theoretic and optimization-based explanations, which addressed this issue to some extent, but not all reviewers found it fully conclusive.

A second concern involved robustness and generality. Reviewers questioned the method’s reliance on model-inversion–based synthetic data and its behavior under low-quality, collapsed, or extreme conditions. Although the rebuttal provided additional ablations, OOD experiments, and architecture-level validation, residual uncertainty remained for some reviewers.

Finally, reviewers noted gaps in experimental scope and positioning. Requests included clearer reporting of computational cost, broader stress testing, and more careful positioning relative to recent data-free or distillation-based defenses. The rebuttal clarified several of these points but did not completely eliminate all concerns.

**Reviewer Concerns:**

Addressed by the Rebuttal:
The rebuttal clarified the intuition and mechanism behind prediction skew with additional theoretical discussion, strengthened empirical evidence through ablations and ImageNet-scale results, and demonstrated robustness across architectures and data sources (including OOD data). It also addressed questions regarding computational cost and practical feasibility, alleviating concerns about scalability and deployment.

Still Outstanding:
Some reviewers remained concerned about the lack of strong theoretical guarantees against fully adaptive attacks, the method’s sensitivity under extreme or degraded synthetic data conditions, and the limited exploration of worst-case or highly adaptive adversaries. In addition, while positioning relative to recent data-free and distillation-based defenses was improved, some ambiguity regarding conceptual novelty persists.

**Reviewer Scores:**

Reviewer g5od:
Likely unchanged or slightly increased. The rebuttal addressed clarification and robustness concerns without introducing new weaknesses, reinforcing the reviewer’s already favorable view.

Reviewer LcoR:
Likely slightly increased. Additional theoretical clarification, expanded experiments (OOD data, architectures), and cost analysis directly addressed their main concerns, though some reservations about theory likely remain.

Reviewer ua98 :
Likely slightly increased. While concerns about theoretical guarantees and adaptive attacks were only partially resolved, the rebuttal meaningfully improved clarity, empirical coverage, and positioning, making the paper stronger than initially assessed.

Reviewer NQnR:
Likely unchanged.

---

### Decision · Program_Chairs · 2026-01-26

Reject